# OPA1 deletion in brown adipose tissue improves thermoregulation and systemic metabolism via FGF21

Renata O Pereira[1]*, Alex Marti[1], Angela Crystal Olvera[1], Satya Murthy Tadinada[1], Sarah Hartwick Bjorkman[1,2], Eric Thomas Weatherford[1], Donald A Morgan[3], Michael Westphal[1], Pooja H Patel[1], Ana Karina Kirby[1], Rana Hewezi[1], William Bùi Trân[1], Luis Miguel García-Peña[1], Rhonda A Souvenir[1], Monika Mittal[1], Christopher M Adams[1], Kamal Rahmouni[1,3], Matthew J Potthoff[1,3], E Dale Abel[1]

[1]Fraternal Order of Eagles Diabetes Research Center and Division of Endocrinology and Metabolism, Roy J. and Lucille A. Carver College of Medicine, University of Iowa, Iowa City, United States; [2]Department of Obstetrics and Gynecology, Reproductive Endocrinology and Infertility, Roy J. and Lucille A. Carver College of Medicine, Iowa City, United States; [3]Department of Neuroscience and Pharmacology, Roy J. and Lucille A. Carver College of Medicine, University of Iowa, Iowa City, United States

*For correspondence:
renata-pereira@uiowa.edu

Competing interests: The authors declare that no competing interests exist.

**Abstract** Adrenergic stimulation of brown adipocytes alters mitochondrial dynamics, including the mitochondrial fusion protein optic atrophy 1 (OPA1). However, direct mechanisms linking OPA1 to brown adipose tissue (BAT) physiology are incompletely understood. We utilized a mouse model of selective OPA1 deletion in BAT (OPA1 BAT KO) to investigate the role of OPA1 in thermogenesis. OPA1 is required for cold-induced activation of thermogenic genes in BAT. Unexpectedly, OPA1 deficiency induced fibroblast growth factor 21 (FGF21) as a BATokine in an activating transcription factor 4 (ATF4)-dependent manner. BAT-derived FGF21 mediates an adaptive response by inducing browning of white adipose tissue, increasing resting metabolic rates, and improving thermoregulation. However, mechanisms independent of FGF21, but dependent on ATF4 induction, promote resistance to diet-induced obesity in OPA1 BAT KO mice. These findings uncover a homeostatic mechanism of BAT-mediated metabolic protection governed in part by an ATF4-FGF21 axis, which is activated independently of BAT thermogenic function.

## Introduction

The prevalence of obesity and its comorbidities, such as type 2 diabetes (T2DM) and cardiovascular disease, has increased dramatically in recent decades (*Arroyo-Johnson et al., 2016*; *Hruby et al., 2015*). Most currently available pharmacological approaches to combat obesity act by reducing caloric intake or impairing fat absorption. However, effective and safe therapies to increase energy expenditure are lacking. The discovery of active brown adipose tissue (BAT) in adult humans has incited interest in exploring BAT activation as a potential strategy for increasing energy expenditure to mitigate obesity and its complications (*Cypess et al., 2015*; *Ruiz et al., 2018*; *Finlin et al., 2020*). Furthermore, BAT has been increasingly recognized as a secretory organ, promoting the release of endocrine factors, or BATokines, that may regulate systemic metabolic homeostasis (*Villarroya et al., 2019*). Therefore, increased understanding of mechanisms regulating BAT thermogenesis and its secretory function could identify new therapeutic strategies for treating obesity and its comorbidities.

Recent studies demonstrated a critical role of mitochondrial dynamics for thermogenic activation of BAT (*Wikstrom et al., 2014*; *Pisani et al., 2018*). Mitochondrial dynamics describes the process by which mitochondria undergo repeated cycles of fusion and fission. It is mediated by several proteins, including the outer mitochondria membrane fusion proteins mitofusins 1 and 2 (Mfn1 and Mfn2), the inner mitochondrial membrane fusion protein optic atrophy 1 (Opa1), and the fission protein dynamin-related protein 1 (Drp1) (*Dorn, 2019*). Norepinephrine treatment induces complete and rapid mitochondrial fragmentation in cultured brown adipocytes through protein kinase A (PKA)-dependent Drp1 phosphorylation, which increases fatty acid oxidation and amplifies thermogenesis (*Wikstrom et al., 2014*; *Pisani et al., 2018*). Mfn2 is also required for BAT thermogenesis, with its absence rendering mice cold-intolerant, but surprisingly resistant to diet-induced insulin resistance (IR) (*Mahdaviani et al., 2017*). An earlier study demonstrated that siRNA-mediated knockdown of OPA1 in brown adipocytes resulted in a modest, but significant, reduction in palmitate oxidation, suggesting a potential role for OPA1 in regulating thermogenesis (*Quirós et al., 2012*). Moreover, indirect evidence in mice lacking the ATP-independent metalloprotease OMA1, which plays an essential role in the proteolytic inactivation of OPA1, supports the notion that OPA1 regulation of fission is important for thermogenesis. Germline OMA1-deficient mice exhibit increased adiposity, decreased energy expenditure, and impaired thermogenesis (*Quirós et al., 2012*). However, given the ubiquitous expression of OPA1 and OMA1, it is impossible to determine from this model the specific contribution of OPA1 to BAT physiology.

In the present study, we investigated the requirement of OPA1 for BAT's adaptation to thermogenic stimuli in vivo by generating mice with BAT-specific ablation of the *Opa1* gene (OPA1 BAT KO mice). Our data demonstrated that lack of OPA1 reduced the activation of the thermogenic gene program in BAT, while surprisingly inducing the expression and secretion of fibroblast growth factor 21 (FGF21) as a BATokine, via an ATF4-dependent mechanism. BAT-derived FGF21 mediates an adaptive response characterized by increased browning of white adipose tissue (WAT), elevated resting metabolic rates, and improved thermoregulation. Nonetheless, FGF21 was dispensable for the resistance to diet-induced obesity (DIO) and IR observed in these mice, which were mediated by alternative mechanisms downstream of ATF4.

## Results

### OPA1 deficiency leads to mitochondrial dysfunction in BAT, while improving energy balance and thermoregulation in mice

To determine the role of OPA1 in BAT physiology, we examined changes in OPA1 levels in BAT in response to high-fat feeding or cold stress. Twelve weeks of high-fat diet (HFD) increased *Opa1* mRNA (*Figure 1A*) and protein levels in BAT (*Figure 1B*). Three days of cold exposure (4°C) induced *Opa1* mRNA in BAT (*Figure 1C*); however, OPA1 total protein levels were significantly reduced (*Figure 1D*). Inner membrane-anchored long OPA1 (L-OPA1) undergoes proteolytic cleavage by OMA1, resulting in short OPA1 (S-OPA1), which promotes mitochondrial fission (*Lee et al., 2017*). Cold temperatures reduced the ratio of L-OPA1 versus the S-OPA1 (*Figure 1D*), indicating increased proteolytic cleavage of OPA1. Mice deficient for OPA1 specifically in BAT (OPA1 BAT KO) were generated to further investigate the requirement of OPA1 for BAT function. *Opa1* mRNA and protein levels were reduced by 10-fold in BAT (*Figure 1E, F*), but were maintained in other tissues such as WAT, liver, and skeletal muscle (*Figure 1—figure supplement 1A–C*). Uncoupling protein 1 (UCP1) protein levels were unchanged in OPA1-deficient BAT under baseline conditions (*Figure 1F*). Histological analysis revealed increased numbers of enlarged unilocular lipid droplets, suggesting whitening of BAT when OPA1 is deleted (*Figure 1G*). Ultrastructurally, mitochondria appeared more fragmented and lamellar cristae structure was disrupted (*Figure 1G*). Isolated mitochondria from OPA1-deficient BAT revealed impaired mitochondrial bioenergetics, exemplified by reduced pyruvate-malate (*Figure 1H*), and palmitoyl-carnitine-dependent oxygen consumption (*Figure 1I*) and ATP synthesis rates (*Figure 1J*). Mitochondrial dysfunction in BAT has been associated with reduced metabolic rates and increased fat accumulation in mice (*Ellis et al., 2010*; *Lee et al., 2015b*). Although we observed no differences in body weight (*Figure 1—figure supplement 1D*), fat mass (*Figure 1—figure supplement 1E*), or lean mass (*Figure 1—figure supplement 1F*) in 4-week-old KO female mice, oxygen consumption was increased at 7 weeks of age (*Figure 1—figure*

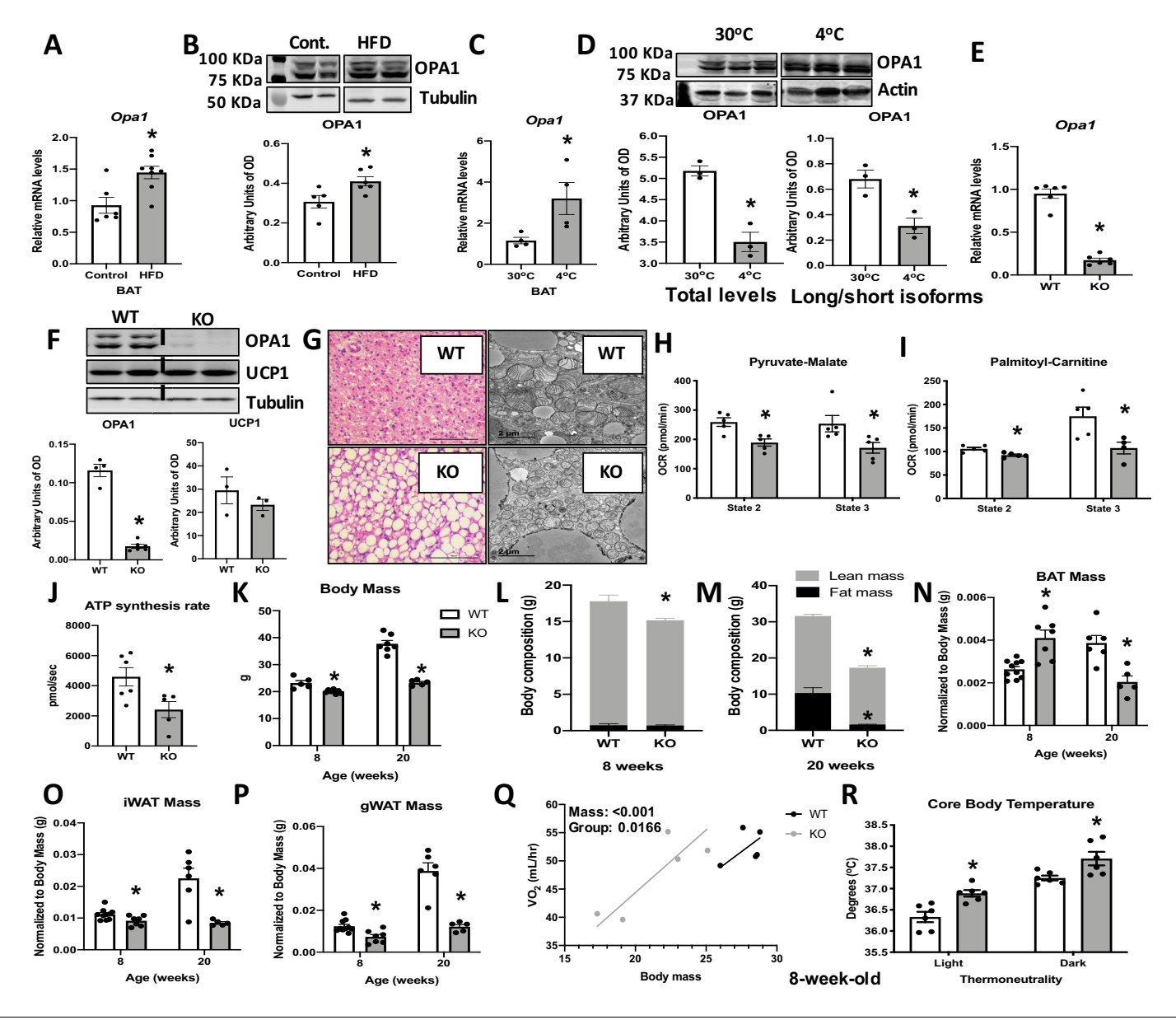

**Figure 1.** Optic atrophy 1 (OPA1) deficiency leads to mitochondrial dysfunction in brown adipose tissue (BAT), while improving energy balance and thermoregulation in mice. (**A, B**) OPA1 expression in BAT of wild-type (WT) mice fed either control (10% fat) or a high-fat diet (HFD 60% fat) for 12 weeks. (**A**) *Opa1* mRNA expression in BAT. (**B**) Representative immunoblot of OPA1 and densitometric analysis of OPA1 normalized by tubulin (images were cropped from the same membrane). (**C, D**) OPA1 expression in BAT of WT mice maintained at 30°C or 4°C for 3 days. (**C**) *Opa1* mRNA expression in BAT. (**D**) Representative immunoblot of OPA1 and densitometric analysis of OPA1 (total levels and long/short isoforms – images were cropped from the same membrane). (**E–J**) Morphological and functional characterization of BAT from 8-week-old OPA1 BAT KO mice (KO). (**E**) *Opa1* mRNA expression in BAT. (**F**) Representative immunoblot in BAT of OPA1 and UCP1 and densitometric analysis normalized to tubulin (dashed line separates genotypes). (**G**) Representative images of H&E-stained histological sections and electron micrographs of BAT from 8-week-old WT and KO mice (n = 3). Scale bar = 100 μm and 2 μm, respectively. (**H, I**) Functional analysis of mitochondria isolated from BAT of WT and KO mice. (**H**) Basal (state 2) and ADP-stimulated (state 3) pyruvate-malate-supported oxygen consumption rates (OCRs). (**I**) State 2 and state 3 palmitoyl-carnitine-supported OCR. (**J**) Palmitoyl-carnitine-supported ATP synthesis rates. (**K–P**) Body mass and body composition in 8- and 20 week-old WT and KO mice. (**K**) Body mass (8 and 20 weeks of age). (**L**) Body composition (8 weeks of age). (**M**) Body composition (20 weeks of age). (**N**) BAT mass. (**O**) Inguinal white adipose tissue (iWAT) mass. (**P**) Gonadal white adipose tissue mass (gWAT). (**Q**) Regression plot comparing oxygen consumption as a function of body mass in mice housed at 30°C. (**R**) Core body temperature in 8-week-old mice housed at 30°C. Data are expressed as means ± SEM. Significant differences were determined by Student's *t*-test, using a significance level of p<0.05. *Significantly different vs. WT mice. VO₂ data was analyzed by ANCOVA. The online version of this article includes the following source data and figure supplement(s) for figure 1:

*Figure 1 continued on next page*

*Figure 1 continued*

**Source data 1.** Optic atrophy 1 (OPA1) deficiency leads to mitochondrial dysfunction in brown adipose tissue (BAT), while improving energy balance and thermoregulation in mice.
**Figure supplement 1.** Age-dependent changes in body composition and glucose homeostasis in optic atrophy 1 (OPA1) brown adipose tissue (BAT) knockout (KO) mice.
**Figure supplement 1—source data 1.** Age-dependent changes in body composition and glucose homeostasis in optic atrophy 1 (OPA1) brown adipose tissue (BAT) knockout (KO) mice.
**Figure supplement 2.** Data collected in 8-week-old optic atrophy 1 (OPA1) brown adipose tissue (BAT) knockout mice (KO) and their wild-type littermate controls (WT) reared at thermoneutrality.
**Figure supplement 2—source data 1.** Data collected in 8-week-old optic atrophy 1 (OPA1) brown adipose tissue (BAT) knockout mice (KO) and their wild-type litter mate controls (WT) reared at thermoneutrality.

*supplement 1G*), before changes in body mass were detected. OPA1 BAT KO male mice exhibited a small, but significant, reduction in body mass, which became more striking with age (*Figure 1K*). Total fat mass was unchanged, and total lean mass was reduced in 8-week-old KO mice (*Figure 1L*), whereas percent fat mass and lean mass relative to body weight were not significantly changed (*Figure 1—figure supplement 1H, I*). However, the expected age-dependent increase in total fat mass and lean mass was strikingly attenuated in 20-week-old KO mice, which had reduced total fat mass and lean mass, relative to wild-type (WT) mice (*Figure 1M*). Percent fat mass was reduced, while percent lean mass was increased in 20-week-old KO mice (*Figure 1—figure supplement 1J, K*). Although fat mass was unchanged in 8-week-old KO mice, BAT mass was significantly increased at this age, but significantly reduced by 20 weeks (*Figure 1N*). In contrast, weight of inguinal (*Figure 1O*) and gonadal (*Figure 1P*) WAT depots (gonadal white adipose tissue [gWAT] and inguinal white adipose tissue [iWAT], respectively) were significantly reduced in KO mice at both 8 and 20 weeks of age. Despite these changes in body composition, glucose tolerance was unaffected in 8-week (*Figure 1—figure supplement 1L, M*) and 20-week-old KO mice (*Figure 1—figure supplement 1N, O*). The reduction in body weight and fat mass persisted in aging mice as 50-week-old female mice exhibited an approximate 40% reduction in body mass (*Figure 1—figure supplement 1P*), a threefold reduction in body fat (*Figure 1—figure supplement 1Q*), and a significant increase in lean mass (*Figure 1—figure supplement 1R*), relative to body weight. At this age, glucose tolerance was significantly improved in KO mice relative to WT mice (*Figure 1—figure supplement 1S, T*). To further elucidate mechanisms responsible for the weight loss, 8-week-old mice were studied in metabolic chambers. At thermoneutrality (30°C), ANCOVA analysis indicated a leftward shift in the relationship between body weight and oxygen consumption in KO mice. Thus, relative to total body mass, KO mice have higher oxygen consumption rates versus WT animals (*Figure 1Q*). This likely contributes to the lean phenotype in KO mice as no changes were detected in food intake (*Figure 1—figure supplement 1U*) or locomotor activity (*Figure 1—figure supplement 1V*) between genotypes. Despite reduced mitochondrial fatty acid oxidation in BAT, core body temperature was significantly increased in KO mice (*Figure 1R*), indicating increased thermogenesis, even at thermoneutrality (30°C). To determine if these phenotypic changes would persist in the absence of lifelong thermogenic activation, a separate cohort of mice was raised at 30°C. Body mass remained reduced in KO mice compared to WT mice (*Figure 1—figure supplement 2A*), while percent fat mass (*Figure 1—figure supplement 2B*) and lean mass (*Figure 1—figure supplement 2C*) were unchanged between genotypes. As observed in mice raised at room temperature (22°C), BAT mass (*Figure 1—figure supplement 2D*) was increased, whereas gWAT (*Figure 1—figure supplement 2E*) and iWAT mass (*Figure 1—figure supplement 2F*) were significantly reduced in KO mice reared at 30°C. Resting metabolic rates remained elevated in KO mice raised at 30°C (*Figure 1—figure supplement 2G*). Food intake was unchanged between WT and KO mice (*Figure 1—figure supplement 2H*), but locomotor activity (*Figure 1—figure supplement 2I*) was significantly reduced in KO mice during the dark cycle. Together, our data strongly suggest that the improved energy balance and thermoregulation observed in OPA1 BAT KO mice occur independently of BAT thermogenic activation.

## OPA1 BAT KO mice exhibit improved tolerance to cold despite impaired thermogenic activation of BAT

Impaired mitochondrial function in BAT has been linked to cold intolerance in mice (*Mahdaviani et al., 2017*; *Quirós et al., 2012*; *Lee et al., 2015b*). To determine the response of OPA1 BAT KO to cold stress, we first measured rectal body temperature during acute cold exposure. Body temperature declined at a slower rate in KO mice relative to WT mice during 4 hr of cold exposure (*Figure 2A*). Next, we prolonged the thermal challenge by exposing a separate cohort of mice to 4°C for 3 days. mRNA analysis revealed increased expression of the thermogenic genes *Ucp1* (*Figure 2B*), *Prdm16* (*Figure 2C*), and *Ppargc1a* (*Figure 2D*) in BAT of WT mice after 3 days of cold exposure, which was significantly attenuated in KO mice. Nonetheless, core body temperature was elevated in KO mice and was significantly increased in the dark cycle (*Figure 2E*). Thus, mice lacking OPA1 in BAT have improved thermoregulatory capacity despite BAT dysfunction. The relationship between whole animal oxygen consumption was shifted to the left in KO mice, indicating that body weight-adjusted oxygen consumption was increased in KO mice, although absolute levels were not different from WT mice after 3 days of cold exposure (*Figure 2F*). The increase in locomotor activity (*Figure 2G*) and food intake (*Figure 2H*) observed in WT mice during the dark cycle were attenuated in KO mice.

## OPA1 deletion in BAT results in compensatory browning of WAT

To determine the presence of compensatory browning of WAT in OPA1 BAT KO mice, we performed morphological and biochemical analysis of the inguinal fat depots (iWAT), which are prone to undergoing browning in mice. Histologically, iWAT morphology in KO mice revealed several regions presenting smaller adipocytes with multilocular lipid droplets that resembled brown adipocytes (*Figure 3A*). Immunohistochemistry (*Figure 3A*) and immunoblots of mitochondrial protein (*Figure 3B*) both revealed significant induction of uncoupling protein 1 (UCP1) in KO mice. Surprisingly, OPA1 protein levels were also increased in mitochondria isolated from iWAT (*Figure 3B*). Because the specific *Ucp1* Cre mouse used in the present study has been shown to promote recombination of floxed alleles in both brown and beige adipocytes, particularly in response to cold (*Wu et al., 2020*; *Kong et al., 2014*), we examined *Cre* expression in iWAT of WT and KO mice, relative to BAT. *Cre* expression in the BAT of KO mice was ~300-fold higher than its expression in iWAT, which was not statistically increased from WT iWAT (*Figure 3—figure supplement 1A*). These data suggest that *Cre* expression in iWAT is insufficient to promote recombination and deletion of OPA1 at ambient temperature conditions in this model. Mitochondria ultrastructure in beige iWAT was characterized by abundant and tightly organized lamellar-cristae in KO mice relative to WT mice (*Figure 3C*). Transcriptionally, mRNA expression of thermogenic and fatty acid oxidation genes (*Figure 3D*) was induced in iWAT of KO mice, which correlated with increased pyruvate-malate (*Figure 3E*) and palmitoyl-carnitine supported oxygen consumption rates (*Figure 3F*) in isolated mitochondria. As reported in other models of BAT dysfunction (*Lynes et al., 2015*; *Schulz et al., 2013*), we observed increased sympathetic nerve activity (SNA) in iWAT of KO mice, as measured by increased protein levels of tyrosine hydroxylase relative to WT mice (*Figure 3G*), and increased efferent inguinal WAT SNA measured directly by nerve recording (*Figure 3H*).

To investigate the role of brown adipokines or 'BATokines' in the adaptations observed in OPA1 BAT KO mice, we measured the mRNA expression of a subset of previously described BATokines. Of these targets, *Fgf21* was highly induced in BAT of KO mice (*Figure 3I*), which correlated with increased circulating concentrations of FGF21 in KO mice under ad libitum-fed conditions (*Figure 3J*) or after a 6 hr fast (*Figure 3—figure supplement 1B*). Notably, *Fgf21* mRNA expression was reduced in the liver (*Figure 3—figure supplement 1C*), suggesting that BAT, rather than liver, contributed to FGF21 circulating levels in KO mice. Short-term knockdown of OPA1 in brown adipocytes (*Figure 3K, L*) was sufficient to increase the release of FGF21 (*Figure 3M*) into the cell culture media, demonstrating that OPA1 deletion induces FGF21 secretion in a cell-autonomous manner. Thus, OPA1 deficiency in brown adipocytes induces secretion of FGF21 independently of thermogenic activation.

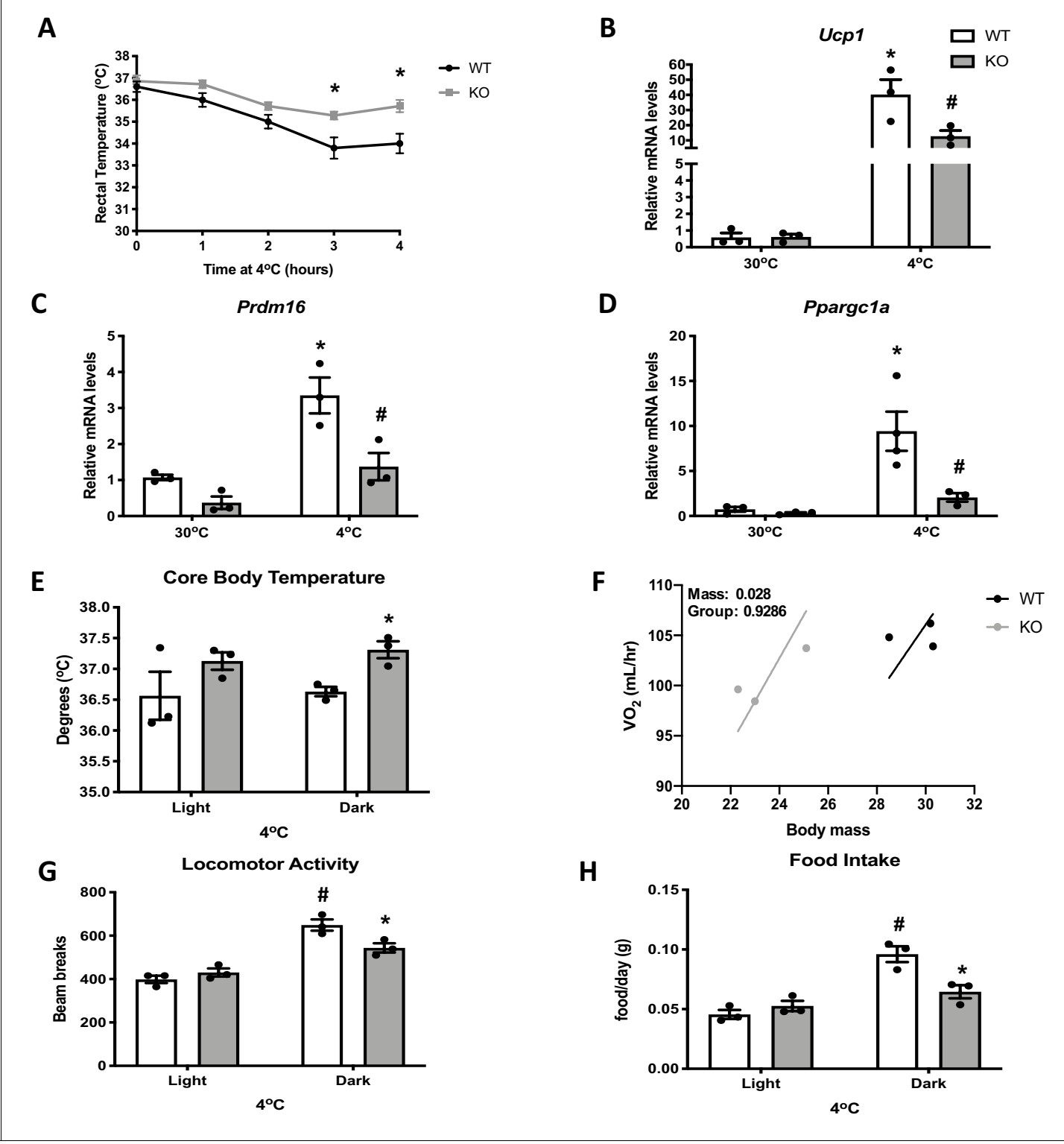

**Figure 2.** Optic atrophy 1 (OPA1) brown adipose tissue (BAT) knockout (KO) mice exhibit improved tolerance to cold despite impaired thermogenic activation of BAT. (A) Rectal temperature in 8-week-old wild-type (WT) and KO mice exposed to acute cold stress (4°C) over the period of 4 hr. (B–D) mRNA expression of thermogenic genes in BAT of WT and KO mice housed at 30°C or 4°C for 3 days. (B) Relative *Ucp1* mRNA levels. (C) Relative *Prdm16* mRNA levels. (D) Relative *Ppargc1α* mRNA levels. mRNA expression was normalized to *Gapdh.* (E, F) Indirect calorimetry and core body temperature in WT and KO mice exposed to 4°C for 3 days. (E) Core body temperature. (F) Regression plot comparing oxygen consumption as a function of body mass in mice housed at 4°C. (G) Locomotor activity. (H) Food intake (average for each cycle). Data are expressed as means ± SEM. *Figure 2 continued on next page*

*Figure 2 continued*

Significant differences were determined by two-Way ANOVA using a significance level of $p < 0.05$. *Significantly different vs. WT mice or vs. 30°C, #significantly different from light cycle or WT mice at 4°C. $VO_2$ data was analyzed by ANCOVA.

The online version of this article includes the following source data for figure 2:

**Source data 1.** Optic atrophy 1 (OPA1) brown adipose tissue (BAT) knockout (KO) mice exhibit improved tolerance to cold despite impaired thermogenic activation of BAT.

## BAT-derived FGF21 is required for increased resting metabolic rates and improved thermoregulation in mice lacking OPA1 in BAT during isocaloric feeding

To determine if BAT-derived FGF21 mediates the systemic metabolic adaptations in OPA1 BAT KO mice, we generated mice lacking both OPA1 and FGF21 specifically in BAT (DKO mice). mRNA expression of *Opa1* and *Fgf21* was significantly reduced in BAT of DKO mice (*Figure 4A*), which completely normalized circulating FGF21 levels (*Figure 4B*). Pyruvate-malate (*Figure 4—figure supplement 1A*) and plamitoyl-carnitine (*Figure 4—figure supplement 1B*) supported state 2 and state 3 mitochondrial respirations were reduced in the BAT of DKO mice to the same extent as observed in OPA1 BAT KO mice. Furthermore, mRNA expression of thermogenic genes was repressed in the BAT of DKO mice (*Figure 4—figure supplement 1C*). Body mass was normalized in DKO mice (*Figure 4C*), and fat mass and lean mass were unchanged (*Figure 4D, E*) between 8-week-old DKO mice and age-matched WT controls. Similar to OPA1 BAT KO mice, BAT mass was increased (*Figure 4F*) in DKO mice at 8 weeks of age, reiterating that FGF21 does not impact the BAT phenotype in this model. In contrast, the reduction in gWAT and iWAT mass observed in OPA1 BAT KO mice was no longer detectable in DKO mice (*Figure 4G, H*). There were no statistically significant changes in oxygen consumption between DKO mice and their WT littermate controls, as analyzed by ANCOVA (*Figure 4I*). Locomotor activity (*Figure 4—figure supplement 1D*) and food consumption (*Figure 4—figure supplement 1E*) were also unchanged between genotypes at 30°C.

The increase in baseline core body temperature observed in OPA1 BAT KO mice was completely lost in DKO mice, suggesting that BAT-derived FGF21 mediates the increase in core body temperature at 30°C (*Figure 4J*). Moreover, there were no statistically significant changes in oxygen consumption between DKO mice and their WT littermate controls at 4°C, as analyzed by ANCOVA (*Figure 4—figure supplement 1F*). Core body temperature tended to be lower after 3 days of cold exposure but did not reach statistical significance (*Figure 4K*). However, the last temperature recorded by telemetry for each mouse was significantly reduced in DKO mice and better reflects the defect in thermoregulation observed in these mice (*Figure 4L*). This was primarily due to a steep reduction in core body temperature in four out of seven DKO mice, which died of hypothermia before the end of the cold exposure studies (data not shown). Locomotor activity was unchanged between genotypes (*Figure 4—figure supplement 1G*), while food intake was significantly reduced in DKO mice during the dark cycle, likely due to hypothermia (*Figure 4—figure supplement 1H*). FGF21 has been implicated in browning of WAT (*Fisher et al., 2012*). We, therefore, measured browning markers in iWAT of DKO mice. UCP1 and OPA1 protein levels (*Figure 4M*) were unchanged in mitochondria isolated from iWAT of DKO mice (*Figure 4M, N*). Of note, although tyrosine hydroxylase protein levels remained elevated in iWAT of DKO mice (*Figure 4O, P*), the induction of thermogenic genes in iWAT observed in OPA1 BAT KO mice was absent in DKO mice (*Figure 4Q*). Furthermore, pyruvate-malate- (*Figure 4R*) and palmitoyl-carnitine-supported (*Figure 4S*) state 2 and state 3 mitochondrial respirations were unchanged between mitochondria isolated from iWAT of WT and DKO mice. Thus, BAT-derived FGF21 is required for the compensatory browning of iWAT and for thermoregulation in OPA1 BAT KO mice fed isocaloric diet.

## OPA1 deletion in BAT prevents DIO and IR

Although impaired BAT thermogenesis is frequently associated with increased susceptibility to DIO (*Feldmann et al., 2009*; *Bachman et al., 2002*; *Lowell et al., 1993*), we hypothesized that the increased resting metabolic rates could protect OPA1 BAT KO mice from DIO. Indeed, KO mice fed HFD for 12 weeks weighed the same as WT mice fed low-fat control diets (*Figure 5A*). KO mice fed the control diet weighed significantly less than WT mice fed the same diet (*Figure 5A*). The

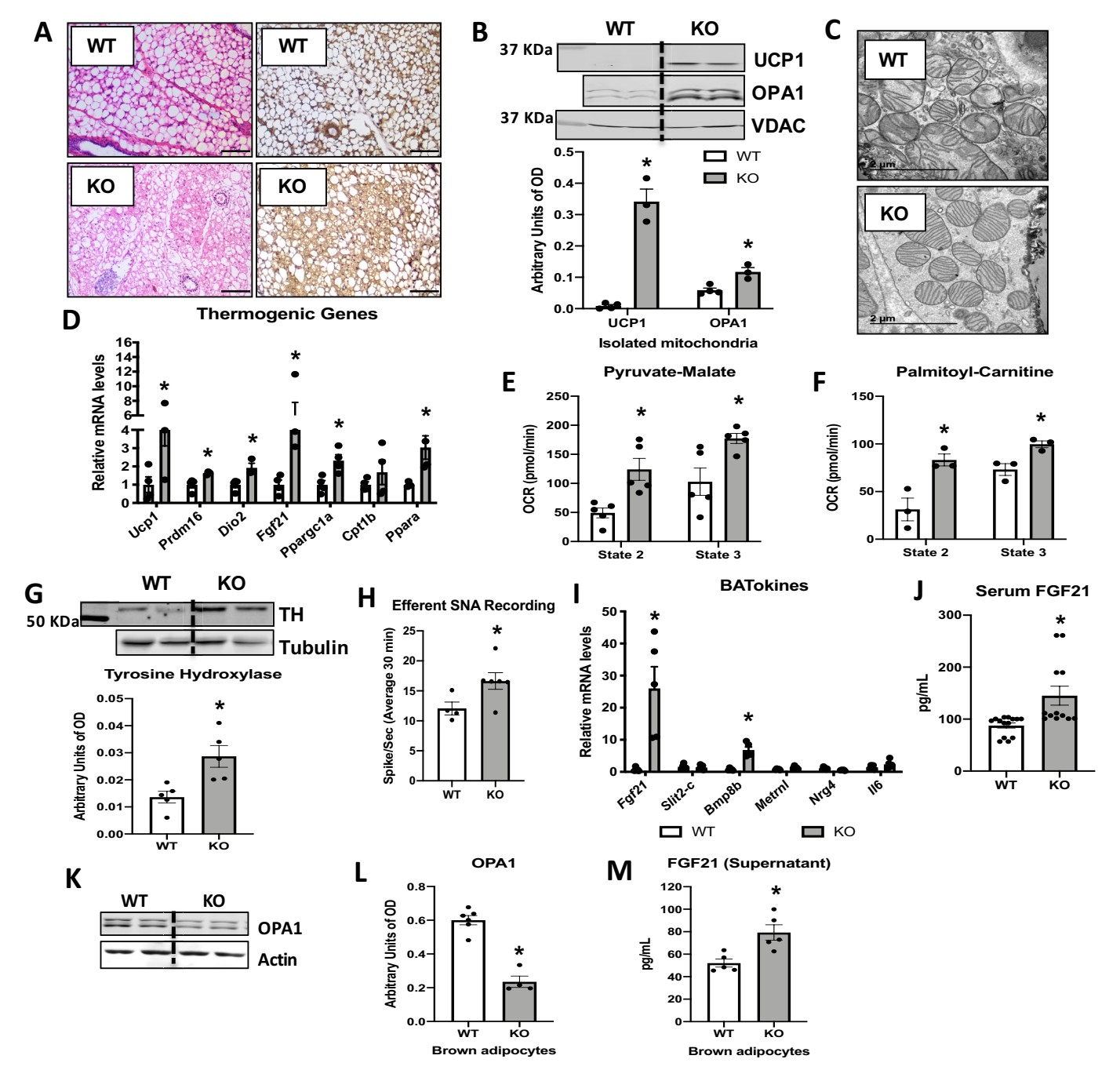

**Figure 3.** Optic atrophy 1 (OPA1) deletion in brown adipose tissue (BAT) results in compensatory browning of white adipose tissue (WAT). (A–G) Morphological and functional characterization of inguinal white adipose tissue (iWAT) in 8-week-old wild-type (WT) and knockout (KO) mice. (A) Representative iWAT sections stained with H&E or after immunohistochemistry against uncoupling protein 1 (UCP1). Scale bar = 100 μm (n = 3). (B) Representative immunoblot (dashed line separates genotypes) and densitometric analysis of UCP1 and OPA1 in mitochondria isolated from iWAT normalized to VDAC. (C) Representative electron micrographs of iWAT from WT and KO mice. Scale bar = 2 μm (n = 3). (D) mRNA expression of thermogenic genes. (E, F) Functional analysis of mitochondria isolated from iWAT. (E) State 2 and state 3 pyruvate-malate-supported mitochondrial oxygen consumption rate (OCR). (F) State 2 and state 3 palmitoyl-carnitine-supported mitochondrial OCR. (G) Representative immunoblot (dashed line separates genotypes) and densitometric analysis of tyrosine hydroxylase (TH) normalized to tubulin. (H) Efferent nerve recording in iWAT. (I) mRNA levels of BATokines in BAT extracts from 8-week-old WT and KO mice. (J) Serum levels of fibroblast growth factor 21 (FGF21) in random-fed 8-week-old WT and KO mice. (K) Representative immunoblots of OPA1 normalized to actin in primary brown adipocytes (dashed line separates genotypes). (L) Densitometric analysis of OPA1 normalized to tubulin in brown adipocytes. (M) FGF21 levels measured in the culture media collected from WT and

*Figure 3 continued on next page*

*Figure 3 continued*

OPA1-deficient brown adipocytes. Data are expressed as means ± SEM. Significant differences were determined by Student's *t*-test, using a significance level of p<0.05. *Significantly different vs. WT mice.

The online version of this article includes the following source data and figure supplement(s) for figure 3:

**Source data 1.** Optic atrophy 1 (OPA1) deletion in brown adipose tissue (BAT) results in compensatory browning of white adipose tissue (WAT).

**Figure supplement 1.** Data collected in 8-week-old optic atrophy 1 (OPA1) brown adipose tissue (BAT) knockout mice (KO) and their wild-type littermate controls (WT) at room temperature conditions.

**Figure supplement 1—source data 1.** Data collected in 8-week-old optic atrophy 1 (OPA1) brown adipose tissue (BAT) knockout KO mice (KO) and their wild-type littermate controls (WT) at room temperature conditions.

---

reduction in body mass occurred at the expense of fat mass as percent fat mass (*Figure 5B*) was significantly reduced and percent lean mass (*Figure 5C*) was significantly increased in KO mice fed HFD. BAT mass (*Figure 5D*), gWAT mass (*Figure 5E*), and iWAT mass (*Figure 5F*) were also significantly reduced in HFD-fed KO mice. ANCOVA analysis revealed increased oxygen consumption in KO mice fed either control or HFD, relative to WT mice fed a control diet, when we controlled for changes in body mass (*Figure 5G*). However, no differences were detected between WT mice fed HFD and WT fed control diet. Food intake (*Figure 5H*) and locomotor activity (*Figure 5I*) were unchanged between genotypes. Thus, increased metabolic rates likely contribute to leanness in these mice. Glucose homeostasis and insulin sensitivity were also improved in KO mice fed HFD relative to WT mice. As expected, glucose tolerance was impaired in HFD-fed WT mice, which was significantly attenuated in KO mice (*Figure 5J, K*). Fasting glucose levels were increased in WT mice, but not in KO mice fed HFD (*Figure 5L*). Similarly, insulin tolerance tests (ITTs) revealed impaired insulin sensitivity in WT mice fed a HFD, which was prevented in KO mice (*Figure 5M, N*). Fasting insulin levels were also significantly reduced in KO mice versus WT mice fed HFD (*Figure 5O*). Hepatic steatosis was attenuated in KO mice (*Figure 5—figure supplement 1A, B*) relative to WT mice fed HFD, and serum triglycerides levels were completely normalized (*Figure 5—figure supplement 1C*). Diet-induced thermogenic activation of BAT was significantly attenuated in KO mice, as evidenced by reduced *Ucp1* mRNA levels (*Figure 5—figure supplement 1D*); however, *Ucp1* transcript levels were significantly elevated in the iWAT of KO mice upon high-fat feeding, indicating increased browning of iWAT (*Figure 5—figure supplement 1E*). Of note, tyrosine hydroxylase levels were significantly reduced in iWAT of KO mice fed HFD, relative to WT mice (*Figure 5—figure supplement 1F*).

## BAT-derived FGF21 does not mediate resistance to DIO in OPA1 BAT KO mice

To determine if the resistance to DIO required BAT-derived FGF21, we fed OPA1/FGF21 BAT DKO mice either a control or a HFD for 12 weeks. Under control diet conditions, DKO mice lacked the reduction in body mass noted in OPA1 KO mice (*Figure 6A*). However, when fed a HFD, the increase in total body mass (*Figure 6A*) and fat mass (*Figure 6B*) observed in WT mice on a HFD was completely prevented in DKO mice. Furthermore, the percent lean mass relative to body mass was reduced in WT mice fed HFD (*Figure 6C*). BAT mass was reduced in DKO mice (*Figure 6—figure supplement 1A*), and the diet-induced increase in gonadal and inguinal WAT mass (*Figure 6—figure supplement 1B, C*) was completely prevented in DKO mice relative to WT controls. Indirect calorimetry confirmed that, in mice fed control diet, oxygen consumption was unchanged in DKO mice relative to WT controls (*Figure 6D*); however, DKO, but not WT mice fed a HFD, had increased oxygen consumption relative to WT mice fed control diet when controlled for changes in body mass (*Figure 6D*). This increase in metabolic rates likely contributed to their resistance to weight gain, as we observed no significant changes in food intake (*Figure 6E*) or locomotor activity (*Figure 6F*) between genotypes, regardless of diet. Furthermore, liver triglyceride levels were significantly reduced in DKO mice relative to WT mice fed a HFD (*Figure 6—figure supplement 1D*), consistent with attenuation of diet-induced hepatic steatosis. BAT-derived FGF21 is also dispensable for the improvements in glucose homeostasis and insulin sensitivity in OPA1 BAT KO mice, as HFD-induced glucose intolerance was attenuated in DKO mice (*Figure 6G, H*), and fasting glucose levels were reduced (*Figure 6I*). Finally, insulin sensitivity was also ameliorated in DKO mice compared to WT fed HFD, as shown by the ITT (*Figure 6J, K*). *Ucp1* mRNA levels were marginally increased in BAT

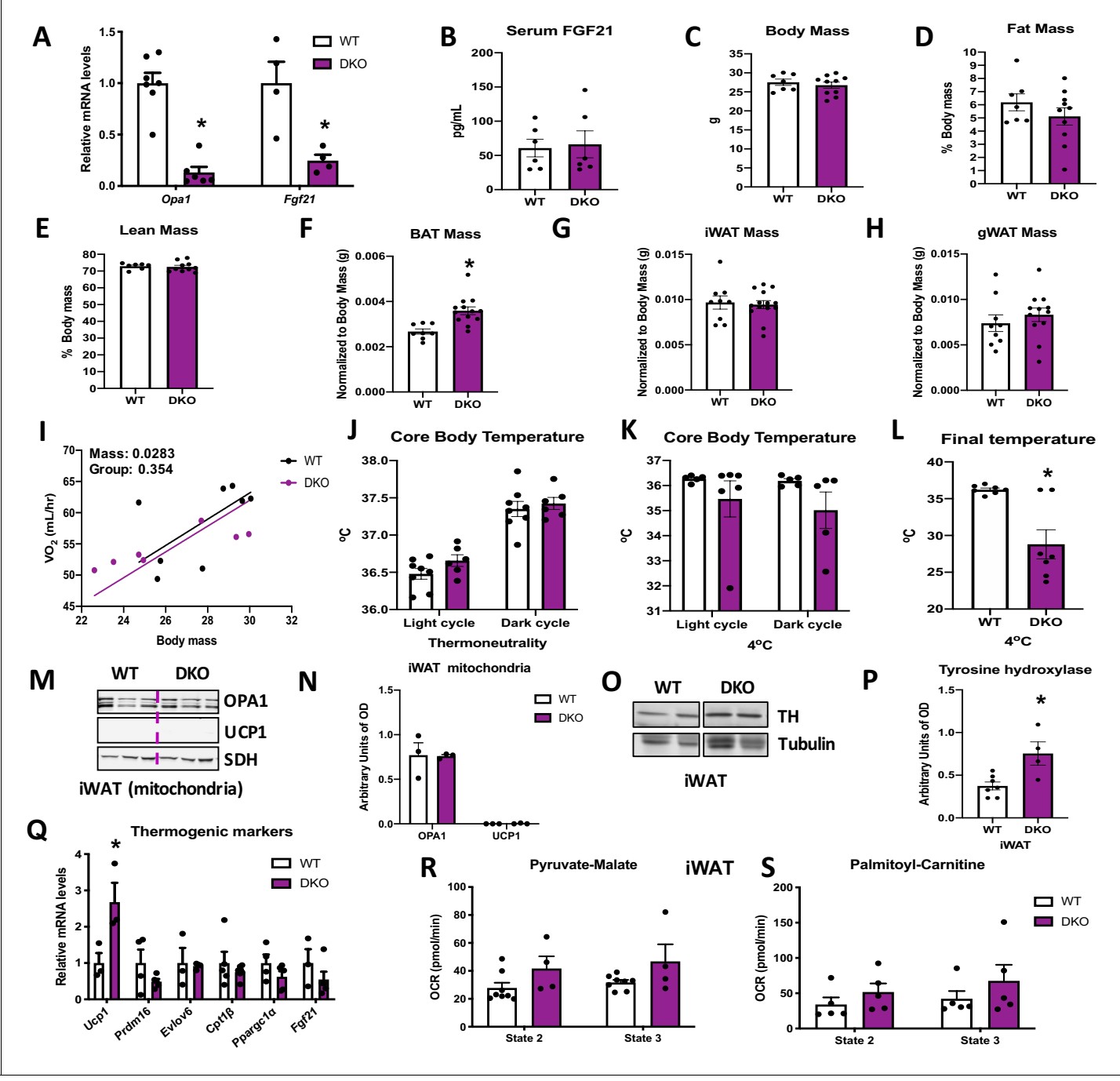

**Figure 4.** Brown adipose tissue (BAT)-derived fibroblast growth factor 21 (FGF21) is required for increased resting metabolic rates and improved thermoregulation in mice lacking optic atrophy 1 (OPA1) in BAT during isocaloric feeding. (A–L) Data characterizing 8–12-week-old OPA1/FGF21 DKO mice. (A) mRNA expression of *Opa1* and *Fgf21* in BAT of DKO mice. (B) FGF21 serum levels collected under ad libitum-fed conditions. (C) Total body mass. (D) Percent fat mass. (E) Percent lean mass. (F) BAT mass. (G) Inguinal white adipose tissue (iWAT) mass. (H) Gonadal white adipose tissue (gWAT) mass. (I) Regression plot comparing oxygen consumption as a function of body mass in mice housed at 30°C. (J) Core body temperature (30° C). (K) Core body temperature (4°C) (data is represented as average core body temperature during the light and dark cycles over 3 days of continuous monitoring). (L) Final core body temperature recorded for each individual mouse (4°C). (M–S) Data of iWAT from 8-week-old DKO mice. (M). Representative immunoblot for OPA1 and uncoupling protein 1 (UCP1) in isolated mitochondria (dashed line separates genotypes). (N) Densitometric analysis of OPA1 and UCP1 protein levels normalized to succinate dehydrogenase (SDH). (O) Representative immunoblot for tyrosine hydroxylase (TH) in iWAT (images were cropped from the same membrane). (P) Densitometric analysis of TH protein levels normalized to tubulin. (Q) mRNA expression of thermogenic genes in iWAT. (R, S) Functional analysis of mitochondria isolated from iWAT. (R) State 2 and state 3 pyruvate-malate-supported mitochondrial OCR. (S) State 2 and state 3 palmitoyl-carnitine-supported mitochondrial oxygen consumption rate (OCR). Data are expressed as

*Figure 4 continued on next page*

Figure 4 continued

means ± SEM. Significant differences were determined by Student's *t*-test or two-way ANOVA, using a significance level of p<0.05. *Significantly different vs. wild-type (WT) mice. VO$_2$ data was analyzed by ANCOVA.

The online version of this article includes the following source data and figure supplement(s) for figure 4:

**Source data 1.** Brown adipose tissue (BAT)-derived fibroblast growth factor 21 (FGF21) is required for increased resting metabolic rates and improved thermoregulation in mice lacking optic atrophy 1 (OPA1) in BAT during isocaloric feeding.

**Figure supplement 1.** Data collected in optic atrophy 1 (OPA1)/fibroblast growth factor 21 (FGF21) brown adipose tissue (BAT) DKO mice and their wild-type littermate controls (WT).

**Figure supplement 1—source data 1.** Data collected in optic atrophy 1 (OPA1)/fibroblast growth factor 21 (FGF21) brown adipose tissue (BAT) DKO mice and their wild-type littermate controls (WT).

---

(*Figure 6—figure supplement 1E*), but substantially increased in iWAT (*Figure 6—figure supplement 1F*) of DKO mice fed a HFD, relative to mice fed control diet, consistent with diet-induced browning and thermogenic activation of iWAT. However, tyrosine hydroxylase protein levels were downregulated in iWAT of DKO mice fed HFD, relative to the WT mice (*Figure 6—figure supplement 1G*). High-fat feeding significantly increased FGF21 serum levels in WT mice, which was prevented in DKO mice (*Figure 6—figure supplement 1H*). Although FGF21 circulating levels were slightly elevated in OPA1 BAT KO mice fed a control diet relative to WT mice, the diet-induced increase in FGF21 levels was blunted in KO mice (*Figure 6—figure supplement 1I*). Thus, FGF21-independent mechanisms mediate the resistance to DIO and IR observed in OPA1 BAT KO mice.

## ATF4 is required for FGF21 induction in OPA1 BAT KO mice

OPA1 deletion in BAT induced endoplasmic reticulum (ER) stress, as demonstrated by increased phosphorylation of the eukaryotic translation initiation factor 2A (eIF2α), which promotes selective translation of the activating transcription factor 4 (ATF4) (*Figure 7A*). Moreover, mRNA expression of the ER stress genes *Atf4*, *Chop,* and *Ern1* was increased in the BAT of KO mice (*Figure 7B*). ATF4 binds to the *Fgf21* promoter to induce its expression in multiple cell types (*Alonge et al., 2017*; *Wan et al., 2014*; *Salminen et al., 2017*). We, therefore, generated mice with concurrent deletion of *Opa1* and *Atf4* in BAT (OPA1/ATF4 BAT DKO) to determine if ATF4 is required for FGF21 induction in KO mice (*Figure 7C, D*). Indeed, *Atf4* deletion in OPA1 BAT KO mice completely normalized *Fgf21* mRNA (*Figure 7E*) and circulating levels (*Figure 7F*). We, then, investigated parameters believed to be regulated by BAT-derived FGF21 in these OPA1/ATF4 BAT DKO mice. Body mass was unchanged in mice lacking ATF4, relative to their WT littermate controls (*Figure 7G*). BAT mass remained elevated in DKO mice (*Figure 7H*), while gonadal (*Figure 7I*) and inguinal (*Figure 7J*) WAT mass were unchanged between genotypes. DKO mice lacked the increase in resting metabolic rates (*Figure 7K*) or core body temperature (*Figure 7L*) observed in OPA1 BAT KO mice at 30°C. Furthermore, activation of thermogenic genes in iWAT was ablated in DKO mice (*Figure 7M*), indicating that the ATF4-FGF21 axis is required for the baseline compensatory browning of iWAT in OPA1 BAT KO mice. At 4°C, core body temperature tended to be lower in DKO mice but did not reach statistical significance (*Figure 7N*). Nonetheless, the last temperature recorded by telemetry for each mouse was significantly reduced in DKO mice (*Figure 7O*). Similar to observations in OPA1/FGF21 DKO mice, a subset of mice died of cold-induced hypothermia, adding variability to the averaged data. Cold-induced thermogenic activation of BAT (*Figure 7P*) and activation of browning were attenuated in DKO mice, as demonstrated by reduced mRNA expression of thermogenic genes (*Figure 7Q*) and reduced UCP1 protein levels (*Figure 7R*). Thus, concomitant deletion of OPA1 and ATF4 in BAT phenocopies the effects of FGF21 deletion in OPA1 BAT KO mice.

## ATF4 induction in BAT is necessary for the resistance to DIO and IR in OPA1 BAT KO mice

To test the role of ATF4 in the protection against DIO and IR observed in OPA1 BAT KO mice, we fed OPA1/ATF4 BAT DKO mice and their respective WT littermate controls HFD for 12 weeks. Deletion of *Atf4* in OPA1 BAT KO mice completely abrogated the resistance to DIO. At the end of 12 weeks of high-fat feeding, body weight (*Figure 8A*) and fat mass (*Figure 8B*) were equivalent between WT and DKO mice. Lean mass was also unchanged between genotypes (*Figure 8C*). Glucose clearance was similarly impaired in WT and DKO mice, as demonstrated by the glucose

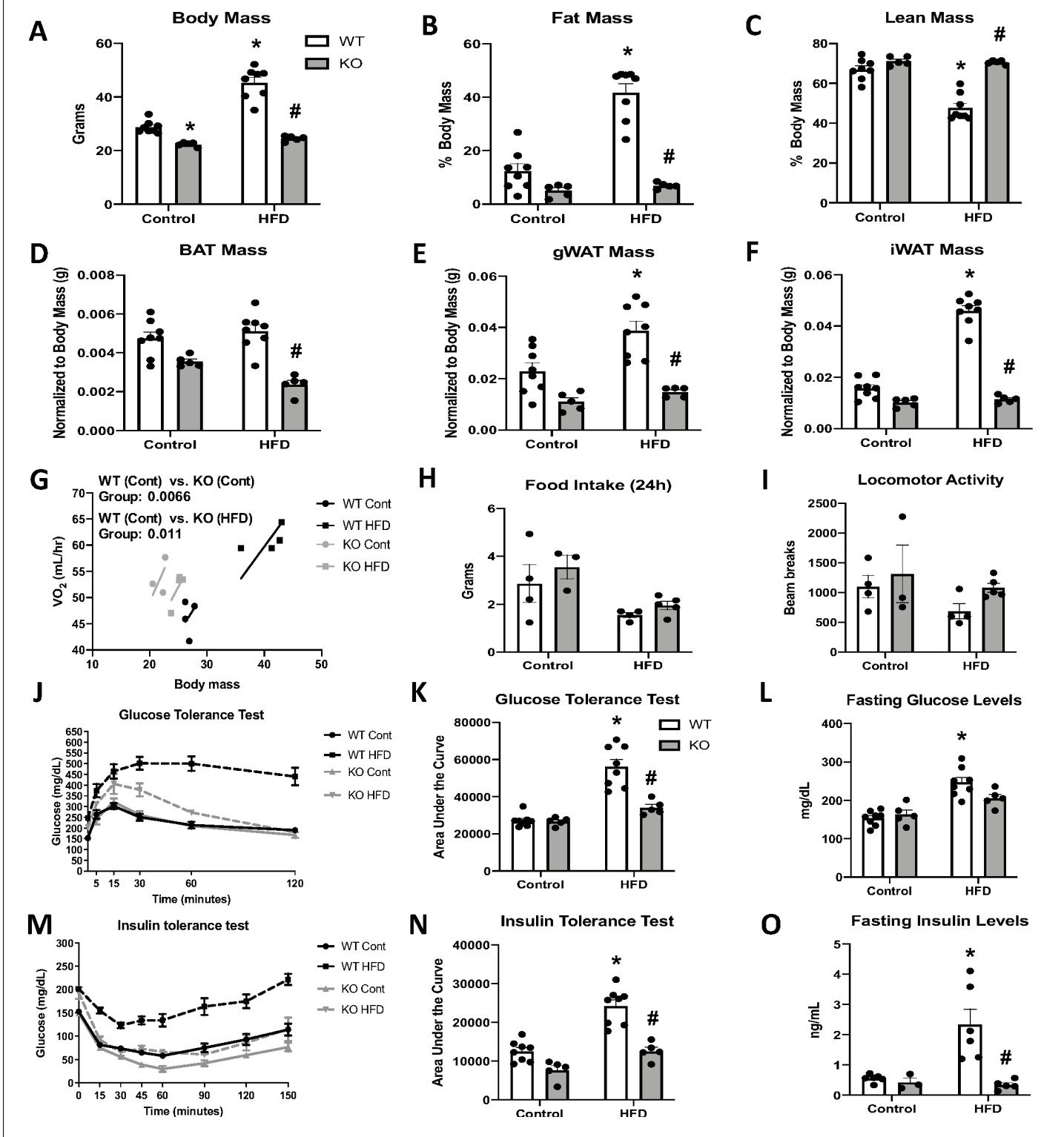

**Figure 5.** Optic atrophy 1 (OPA1) deletion in brown adipose tissue (BAT) prevents diet-induced obesity and insulin resistance. (A–O) Data from wild-type (WT) and OPA1 BAT knockout (KO) mice fed either a control diet (Cont) or a high-fat diet (HFD) for 12 weeks. (A) Total body mass. (B) Percent ratio of fat mass to body mass. (C) Percent ratio of lean mass to body mass. (D) BAT mass. (E) Gonadal white adipose tissue (gWAT) mass. (F) Inguinal white adipose tissue (iWAT) mass. (G) Regression plot comparing oxygen consumption as a function of body mass. (H) Food intake during a 24 hr period. (I) Locomotor activity. (J) Glucose tolerance test (GTT). (K) Area under the curve for the GTT. (L) Fasting glucose levels. (M) Insulin tolerance test (ITT). (N) Area under the curve for the ITT. (O) Fasting insulin levels. Data are expressed as means ± SEM. Significant differences were determined

*Figure 5 continued on next page*

*Figure 5 continued*

by two-way ANOVA, using a significance level of p<0.05. *Significantly different vs. WT control, #significantly different vs. WT HFD. VO₂ data was analyzed by ANCOVA.

The online version of this article includes the following source data and figure supplement(s) for figure 5:

**Source data 1.** Optic atrophy 1 (OPA1) deletion in brown adipose tissue (BAT) prevents diet-induced obesity and insulin resistance.

**Figure supplement 1.** Data collected in optic atrophy 1 (OPA1) brown adipose tissue (BAT) knockout mice (KO) and their wild-type littermate controls (WT) fed either control (10% fat content) or high-fat diet (HFD) (60% fat content) for 12 weeks.

**Figure supplement 1—source data 1.** Data collected in optic atrophy 1 (OPA1) brown adipose tissue (BAT) knockout KO mice (KO) and their wild-type litter mate controls (WT) fed either control (10% fat content) or high-fat diet (HFD) (60% fat content) for 12 weeks.

tolerance test (GTT) (*Figure 8D, E*) and fasting glucose levels (*Figure 8F*). Diet-induced IR was similarly induced in both genotypes, as indicated by overlapping ITTs (*Figure 8G, H*) and unchanged fasting insulin levels (*Figure 8I*). The protection against diet-induced hepatic steatosis present in OPA1 BAT KO and OPA1/FGF21 BAT DKO mice was no longer observable in the absence of ATF4 induction in OPA1 BAT KO mice (*Figure 8J, K*).

## Discussion

Mitochondrial fission contributes to BAT thermogenesis (*Wikstrom et al., 2014*; *Pisani et al., 2018*). However, the role of OPA1 in BAT function was not well-understood. We provide direct evidence that OPA1 maintains mitochondrial respiratory capacity and is required for cold-induced activation of the thermogenic gene program in BAT. Mitochondrial fatty acid β-oxidation (FAO) is critical for maintaining the brown adipocyte phenotype both during times of activation and quiescence. FAO also fuels the increase in uncoupled mitochondrial respiration and contributes to inducing the expression of thermogenic genes such as *Ucp1*, *Ppargc1a*, and *Dio2* in response to adrenergic stimulation (*Fedorenko et al., 2012*; *Gonzalez-Hurtado et al., 2018*). Consequently, mice with adipose-specific deficits in FAO are severely cold-intolerant, demonstrating its role in cold-induced thermogenesis (*Ellis et al., 2010*; *Lee et al., 2015b*). Although not directly tested, it is likely that reduced FAO in OPA1 BAT KO mice (KO) contributed to the impaired thermogenic activation of BAT. However, in contrast to many models of mitochondrial dysfunction or FAO defects in BAT, KO mice displayed improved cold adaptation.

The increased core body temperatures at thermoneutrality and heightened tolerance to cold at 4°C correlated with a significant increase in compensatory browning of subcutaneous WAT in KO mice. Several studies reported increased browning of WAT following BAT impairment. Surgical removal of interscapular BAT (iBAT) enhanced WAT browning due to increased sympathetic input to WAT in mice, which was accompanied by reduced adiposity (*Piao et al., 2018*). In addition, denervation of iBAT increased sympathetic input to subcutaneous fat to induce compensatory browning (*Schulz et al., 2013*). Genetic models of BAT paucity demonstrated a similar phenomenon. Ablation of type 1A BMP-receptor (*Bmpr1A*) in brown adipogenic progenitor cells induces a severe paucity of BAT, which increased sympathetic input to WAT, thereby promoting browning, and maintaining normal temperature homeostasis and resistance to DIO (*Schulz et al., 2013*). Lastly, ablating insulin receptor in the Myf5 lineage reduced iBAT mass in mice that maintained normal thermogenesis on the basis of compensatory browning of subcutaneous WAT and increased lipolytic activity in BAT (*Lynes et al., 2015*). We, therefore, postulated that OPA1 deletion in BAT increased sympathetic input to WAT to promote compensatory browning. Indeed, we observed increased tyrosine hydroxylase protein levels and increased efferent SNA in WAT, even at room temperature conditions, suggesting increased sympathetic innervation of subcutaneous fat. This compensatory browning likely contributes to increased core body temperature at both 30°C and at 4°C.

The mechanisms linking browning of WAT when BAT function is impaired are incompletely understood. Our data in KO mice demonstrated that FGF21 secretion from BAT is independent of BAT thermogenic activation and mediates this adaptation. FGF21 release from activated BAT might contribute to many of the metabolic benefits associated with BAT activity (*Giralt et al., 2015*). Furthermore, FGF21 is a key regulator of WAT browning in mice, leading to increased thermogenesis and energy expenditure (*Fisher et al., 2012*). Our data in mice lacking OPA1 and FGF21 in BAT (OPA1/FGF21 DKO) revealed that BAT-derived FGF21 is required for increased browning of WAT in KO

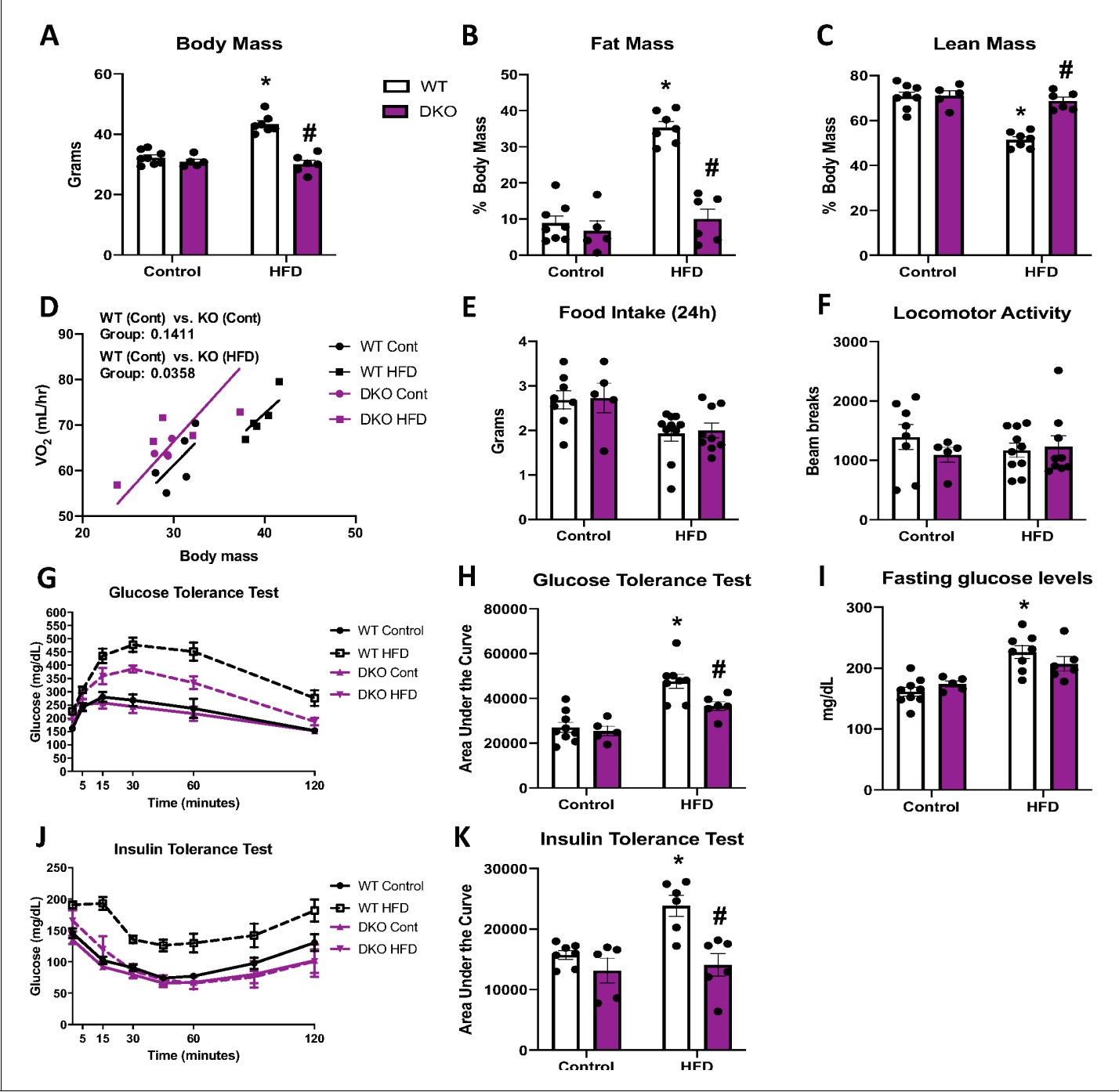

**Figure 6.** Brown adipose tissue (BAT)-derived fibroblast growth factor 21 (FGF21) does not mediate resistance to diet-induced obesity in optic atrophy 1 (OPA1) BAT knockout (KO) mice. (A–K) Data from wild-type (WT) and OPA1/FGF21 DKO mice fed either a control diet (Cont) or a high-fat diet (HFD) for 12 weeks. (A) Total body mass. (B) Percent ratio of fat mass to body mass. (C) Percent ratio of lean mass to body mass. (D) Regression plot comparing oxygen consumption as a function of body mass. (E) Food intake during a 24 hr period. (F) Locomotor activity. (G) Glucose tolerance test. (H) Area under the curve for the glucose tolerance test. (I) Fasting glucose levels. (J) Insulin tolerance test. (K) Area under the curve for the insulin tolerance test. Data are expressed as means ± SEM. Significant differences were determined by two-way ANOVA, using a significance level of $p < 0.05$. *Significantly different vs. WT control, #significantly different vs. WT HFD. $VO_2$ data was analyzed by ANCOVA.

The online version of this article includes the following source data and figure supplement(s) for figure 6:

**Source data 1.** Brown adipose tissue (BAT)-derived fibroblast growth factor 21 (FGF21) does not mediate resistance to diet-induced obesity in optic atrophy 1 (OPA1) BAT knockout (KO) mice.

*Figure 6 continued on next page*

*Figure 6 continued*

**Figure supplement 1.** Data collected in optic atrophy 1 (OPA1) brown adipose tissue (BAT) knockout mice (KO), OPA1/fibroblast growth factor 21 (FGF21) DKO mice or their respective wild-type littermate controls (WT) fed either control (10% fat content) or high-fat diet (HFD) (60% fat content) for 12 weeks.

**Figure supplement 1—source data 1.** Data collected in optic atrophy 1 (OPA1) brown adipose tissue (BAT) knockout mice (KO), OPA1/fibroblast growth factor 21 (FGF21) DKO mice or their respective wild-type littermate controls (WT) fed either control (10% fat content) or high-fat diet (HFD) (60% fat content) for 12 weeks.

mice under isocaloric conditions. These data support the idea that FGF21 is a potent inducer of WAT browning and demonstrates that BAT-derived FGF21 mediates compensatory browning of WAT following OPA1 deletion in BAT, which contributes to the metabolic phenotype of these mice, despite absence of BAT activation.

Pharmacological administration of FGF21 increases energy expenditure and thermogenic gene expression in BAT and WAT (*Fisher et al., 2012*; *Emanuelli et al., 2014*). However, the role of endogenous BAT-derived FGF21 upon cold exposure remains incompletely understood. In KO mice, BAT-derived FGF21 was required for the baseline increase in core body temperature and for the resistance to cold stress. This result contrasts with a recent report demonstrating that FGF21 plays a negligible role in the systemic adaptations to long-term cold exposure in mice, including browning of subcutaneous WAT, in a global FGF21 knockout model (*Keipert et al., 2017*). Conversely, *Fisher et al., 2012* showed that whole-body ablation of FGF21 impaired the response to cold stress when placing mice from 27°C to 5°C for 3 days. These studies utilized global knockout models; thus, the source of FGF21 is unclear. Moreover, liver-derived, but not adipose tissue-derived, FGF21 was shown to enter the circulation within the first hours of cold exposure, contributing to thermoregulation, via its action in the central nervous system (*Ameka et al., 2019*). Together, these studies suggest that BAT-derived FGF21 could be dispensable for thermoregulation during short- and long-term cold acclimation in WT mice. However, our data clearly demonstrates that BAT-derived FGF21 does mediate the compensatory response of mice lacking OPA1 in BAT after short-term cold exposure as OPA1/FGF21 DKO mice undergo a steep decline in core body temperature when exposed to cold. Our model results from constitutive deletion of the *Opa1* gene. Therefore, our phenotype could reflect the effects of long-term exposure to persistent mildly increased endogenous FGF21 circulating levels, leading to considerable browning of WAT even at ambient temperature conditions. This augmented thermogenic response in WAT could prime these animals to better adapt to cold temperatures and might also contribute to the increased resting metabolic rates and leanness in KO mice.

The mechanisms governing FGF21-mediated browning are incompletely understood. FGF21 may directly promote browning of WAT in part, by induction of *Ppargc1a* (*Fisher et al., 2012*), and contributes to the induction of WAT browning during cold acclimation (*Piao et al., 2018*). However, central or peripherally administered FGF21 failed to induce beige fat in mice lacking β-adrenoceptors, indicating the requirement for an intact adrenergic system (*Owen et al., 2014*; *Douris et al., 2015*). Lastly, liver-derived FGF21 mediates thermoregulation via its action in the central nervous system, rather than in adipose tissue (*Ameka et al., 2019*). Together, these data suggest that FGF21 might signal centrally to activate the sympathetic nervous system and promote browning in KO mice. However, we observed increased tyrosine hydroxylase levels in WAT of OPA1/FGF21 DKO mice, which lacked the compensatory browning of WAT. This suggests that, in our model, BAT-derived FGF21 does not mediate the increase in sympathetic innervation of WAT in KO mice, and that increased sympathetic input to WAT is not sufficient to promote browning in the absence of elevated circulating FGF21 levels in mice. Thus, FGF21-independent mechanisms may mediate the compensatory increase in sympathetic tone to WAT in KO mice. However, FGF21 and the sympathetic nervous system may act cooperatively to induce browning in KO mice fed isocaloric diet. Finally, tyrosine hydroxylase levels were reduced in iWAT of both KO and OPA1/FGF21 DKO mice fed HFD, while UCP1 levels were significantly elevated, suggesting that increased sympathetic innervation is dispensable for WAT browning and for the resistance to DIO in KO and DKO mice fed HFD. Although increased tyrosine hydroxylase activity correlated with direct measurements of increased SNA in BAT OPA1 KO mice on normal chow, we did not directly measure SNA in the DIO studies. This more robust measurement of SNA will be required to further elucidate the role, if any,

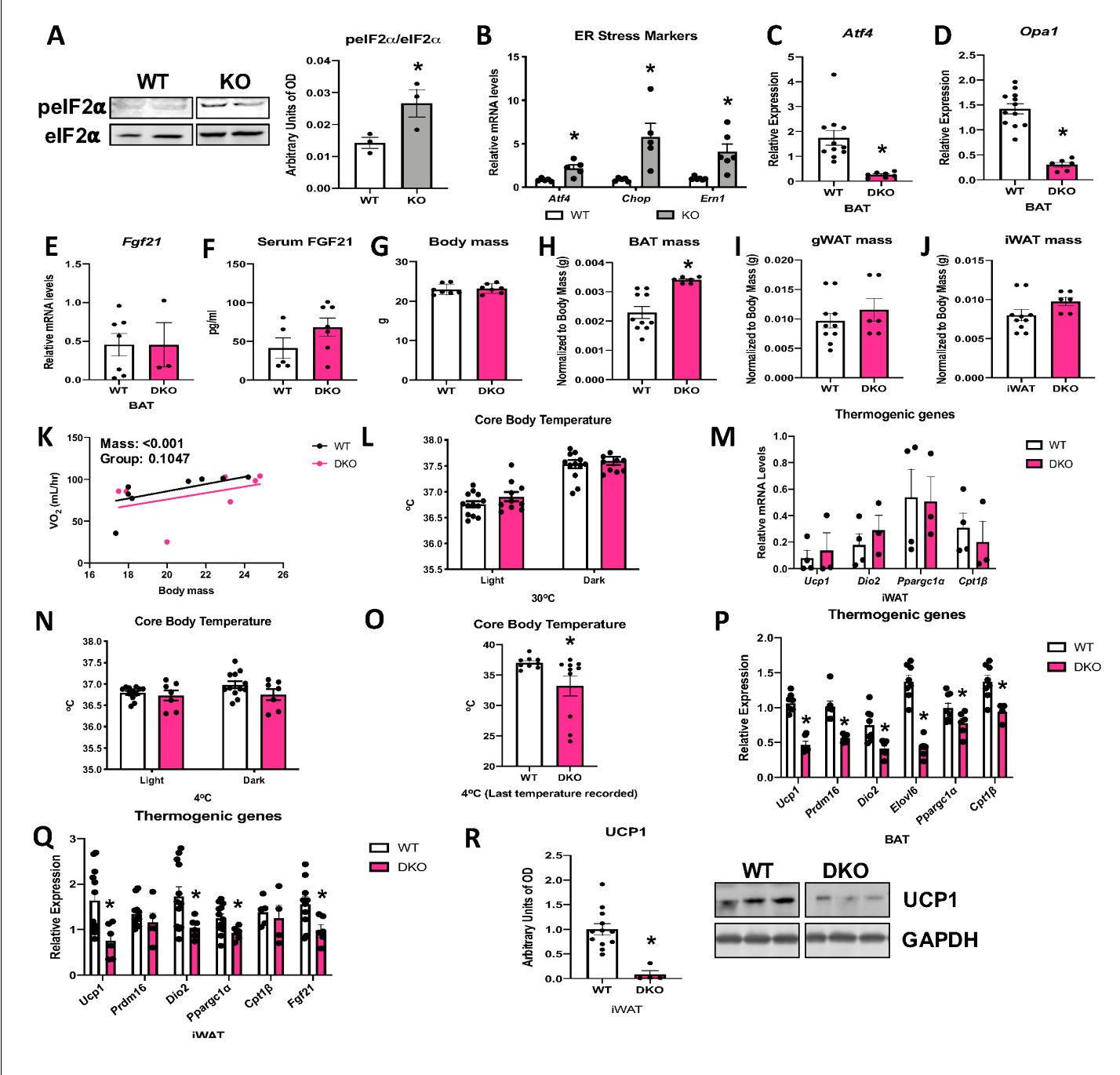

**Figure 7.** Activating transcription factor 4 (ATF4) is required for fibroblast growth factor 21 (FGF21) induction in optic atrophy 1 (OPA1) brown adipose tissue (BAT) knockout (KO) mice. (**A, B**) Analysis of endoplasmic reticulum (ER) stress in BAT tissue from wild-type (WT) and OPA1 BAT KO mice (KO). (**A**) Representative immunoblot for phosphorylated eukaryotic translation initiation factor 2A (eIF2α) over total eIF2α and respective densitometric quantification (images were cropped from the same membrane). (**B**) mRNA expression of ER stress markers. (**C–R**) Data collected in 8–10-week-old OPA1/ATF4 BAT DKO mice. (**C**) mRNA expression of *Atf4* in BAT. (**D**) mRNA expression of *Opa1* in BAT. (**E**) *Fgf21* mRNA expression in BAT. (**F**) FGF21 serum levels at ambient temperature and ad libitum-fed conditions. (**G**) Body mass. (**H**) BAT mass normalized to body mass. (**I**) Gonadal white adipose tissue (gWAT) mass normalized to body mass. (**J**) Inguinal white adipose tissue (iWAT) mass normalized to body mass. (**K**) Regression plot comparing oxygen consumption as a function of body mass in mice housed at 30℃. (**L**) Core body temperature measured at 30℃. (**M**) mRNA expression of thermogenic genes in iWAT samples collected at ambient temperature. (**N**) Core body temperature in DKO mice exposed to 4℃ (data is represented as average core body temperature during the light and dark cycles over 3 days of continuous monitoring). (**O**) Final core body temperature recorded for each individual mouse (4℃). (**P**) mRNA expression of thermogenic genes in BAT samples. (**Q**) mRNA expression of thermogenic genes in iWAT samples. (**R**) Representative immunoblot for uncoupling protein 1 (UCP1) normalized to GAPDH (images were cropped from the same membrane) in

*Figure 7 continued on next page*

*Figure 7 continued*

iWAT and respective densitometric quantification (**P–R** collected after 3 days at 4˚C). Data are expressed as means ± SEM. Significant differences were determined by Student's *t*-test, using a significance level of p<0.05. *Significantly different vs. WT. VO$_2$ data was analyzed by ANCOVA.

The online version of this article includes the following source data for figure 7:

**Source data 1.** Activating transcription factor 4 (ATF4) is required for fibroblast growth factor 21 (FGF21) induction in optic atrophy 1 (OPA1) brown adipose tissue (BAT) knockout (KO) mice.

of sympathetic nervous system activation in the resistance to DIO observed in KO and OPA1/FGF21 DKO mice.

FGF21 is strongly induced in BAT in response to thermogenic stimulation via mechanisms that involve β-adrenergic signaling activation (*Fisher et al., 2012*). Because BAT thermogenic function was impaired in KO mice, and FGF21 levels were elevated even at ambient temperatures conditions, we hypothesized that alternative signaling pathways regulated FGF21 induction in KO mice. We and others have shown that OPA1 deletion in muscle leads to *Fgf21* induction via activation of the ER stress pathway (*Pereira et al., 2017*; *Tezze et al., 2017*). ATF4, a transcription factor downstream of the unfolded protein response (UPR), has been proposed to induce transcriptional regulation of *Fgf21* in models of mitochondrial stress as part of the integrated stress response (*Lee, 2015a*). Our data in mice lacking both OPA1 and ATF4 selectively in BAT demonstrated that ATF4 is required for FGF21 induction in and secretion from BAT in response to OPA1 deletion. Accordingly, lack of ATF4 in BAT recapitulated the effects of FGF21 deficiency in OPA1 BAT KO mice, including lack of baseline browning, normalized metabolic rates, and impaired adaptive thermogenesis. Taken together, our data strongly suggest that ATF4, likely downstream of ER stress activation, is required for FGF21 induction in OPA1 BAT KO mice. Interestingly, a recent study demonstrated induction of ER stress and ATF4 in BAT in response to cold stress (*Flicker et al., 2019*). Future studies focusing on the role of ATF4 in BAT thermogenesis and BATokine secretion might identify mechanisms for BAT-mediated metabolic adaptations that might be independent of β-adrenergic stimulation of BAT.

Surprisingly, BAT-derived FGF21 does not appear to mediate the resistance to DIO and IR in KO mice. These data suggest that factors other than FGF21 may contribute to the lean phenotype in these mice and may mediate the increase in metabolic rates and browning of WAT when mice are fed HFD. Indeed, the diet-induced increase in FGF21 circulating levels was completely blunted in OPA1 BAT KO mice. This finding further supports the conclusion that FGF21 is not required for the systemic adaptations observed when OPA1 BAT KO mice are fed an obesogenic diet. Nonetheless, ATF4 is required for the resistance to DIO as OPA1/ATF4 BAT DKO mice become as obese and insulin resistant as WT mice after high-fat feeding. Future studies investigating the BAT secretome in this mouse model might identify additional BAT-derived secreted factors, which could potentially mediate the resistance to DIO. It is also possible that alternative mechanisms downstream of ATF4 activation, independent of the BAT secretome, could mediate this metabolic protection in OPA1 BAT KO mice.

In conclusion, these studies reveal an important stress response pathway in BAT in the absence of OPA1, consisting of induction of FGF21 as a BATokine via ATF4-dependent mechanisms, which promote leanness and improve thermoregulation when BAT function is dampened. Although FGF21 seems dispensable for the resistance to DIO in OPA1 BAT KO mice, mechanisms downstream of ATF4 are required for this protection. Determining the relevance of this ATF4-FGF21 axis in BAT physiology and BAT-mediated metabolic adaptations may lead to novel therapeutic approaches to combat obesity and associated disorders.

## Materials and methods

**Key resources table**

| Reagent type (species) or resource | Designation | Source or reference | Identifiers | Additional information |
|---|---|---|---|---|

*Continued on next page*

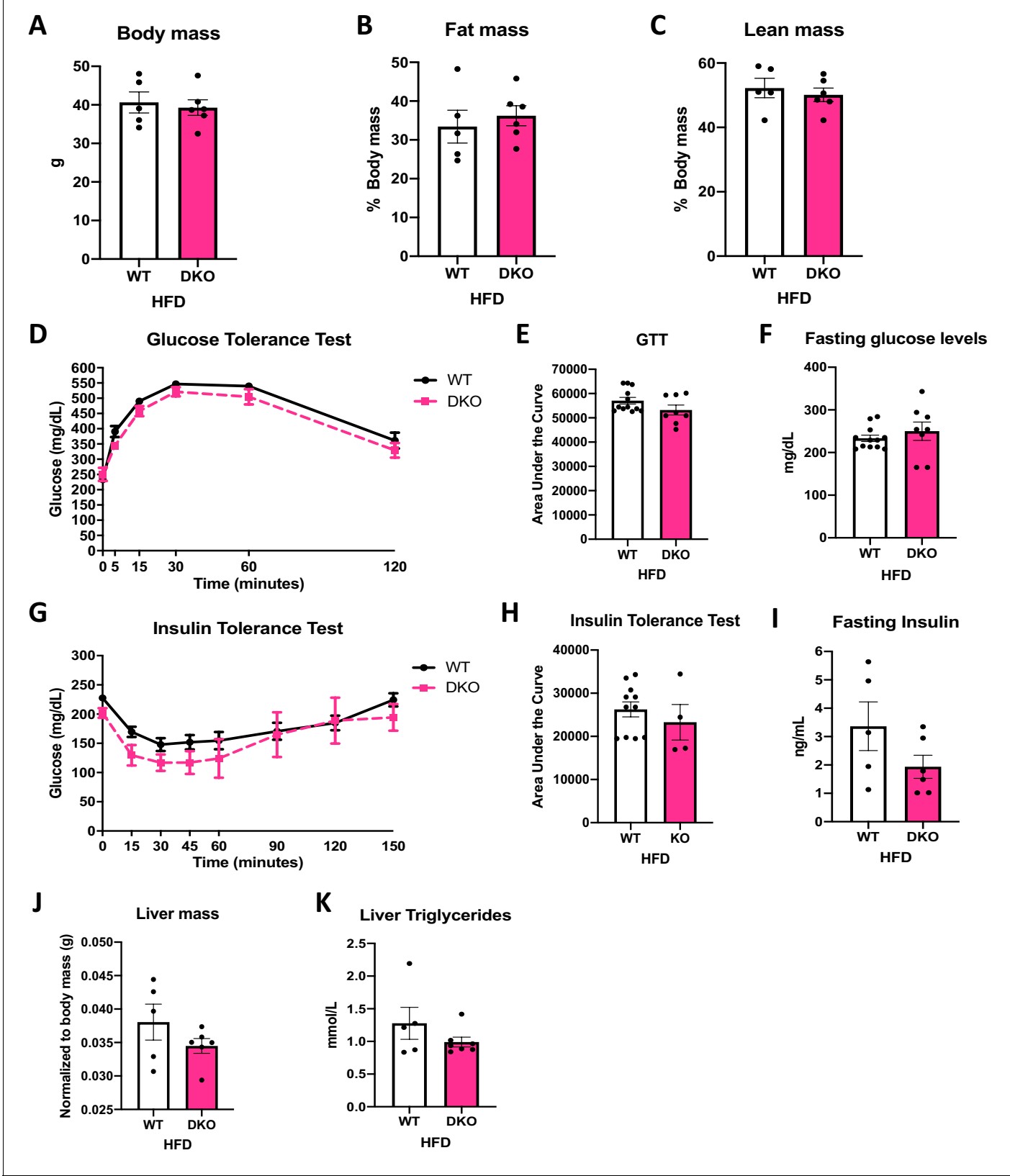

**Figure 8.** Activating transcription factor 4 (ATF4) induction in brown adipose tissue (BAT) is necessary for the resistance to diet-induced obesity (DIO) and insulin resistance (IR) in optic atrophy 1 (OPA1) BAT knockout (KO) mice. (A–K) Data from wild-type (WT) and OPA1/ATF4 BAT DKO mice fed a

*Figure 8 continued on next page*

*Figure 8 continued*

high-fat diet (HFD) for 12 weeks. (A) Total body mass. (B) Percent ratio of fat mass to body mass. (C) Percent ratio of lean mass to body mass. (D) Glucose tolerance test. (E) Area under the curve for the glucose tolerance test. (F) Fasting glucose levels. (G) Insulin tolerance test. (H) Area under the curve for the insulin tolerance test. (I) Fasting insulin levels. (J) Liver mass normalized to body mass. (K) Liver triglycerides levels. Data are expressed as means ± SEM. Significant differences were determined by Student's *t*-test, using a significance level of $p < 0.05$.

The online version of this article includes the following source data for figure 8:

**Source data 1.** Activating transcription factor 4 (ATF4) induction in brown adipose tissue (BAT) is necessary for the resistance to diet-induced obesity (DIO) and insulin resistance (IR) in optic atrophy 1 (OPA1) BAT knockout (KO) mice.

*Continued*

| Reagent type (species) or resource | Designation | Source or reference | Identifiers | Additional information |
|---|---|---|---|---|
| Strain, strain background (mouse, C57Bl/6J) | Murine models | Jackson Laboratories | JAX Stock #024670 RRID:IMSR_JAX:024670 | Tg (Ucp1-cre)1Evdr; male and female |
| Strain, strain background (mouse, C57Bl/6J) | Murine models | Jackson Laboratories | JAX Stock #025124 RRID:IMSR_JAX:025124 | C57BL/6-Tg(Adipoq-cre/ERT2)1Soff/J; male and female |
| Antibody | Anti-OPA1 (Mouse monoclonal) | BD Biosciences | #612606 RRID:AB_399888 | WB (1:1000), primary |
| Antibody | Anti-FGF21 (Rabbit monoclonal) | Abcam | #ab171941 | WB (1:1000), primary |
| Antibody | Anti-GAPDH (Rabbit monoclonal) | Cell Signaling Technology | #2118 RRID:AB_561053 | WB (1:1000), primary |
| Antibody | Anti-VDAC (Rabbit polyclonal) | Thermo Scientific | #PA1-954A RRID:AB_2304154 | WB (1:1000), primary |
| Antibody | Anti-UCP1 (Rabbit polyclonal) | Abcam | #Ab10983 RRID:AB_2241462 | WB (1:1000), primary Histology 1:250 |
| Antibody | Anti-SDH (Mouse monoclonal) | Abcam | #Ab14714 | WB (1:1000), primary |
| Antibody | Anti-α-tubulin (Mouse monoclonal) | Sigma | #T9026 | WB (1:1000), primary |
| Antibody | Anti-β-actin (Rabbit polyclonal) | Sigma | #A2066 RRID:AB_476693 | WB (1:1000), primary |
| Antibody | Anti-tyrosine hydroxylase (Rabbit polyclonal) | Cell Signaling Technology | #2792 RRID:AB_2303165 | WB (1:1000), primary |
| Antibody | Anti-phosphorylated eIF2α serine 51 (Rabbit monoclonal) | Cell Signaling Technology | #3597 | WB (1:1000), primary |
| Antibody | anti-eIF2α (Mouse monoclonal) | Santa Cruz Biotechnology | #SC81261 | WB (1:1000), primary |
| Antibody | IRDye 800CW anti-mouse | LI-COR | #925-32212 RRID:AB_2716622 | WB (1:10,000), secondary |
| Antibody | Alexa Fluor anti-rabbit 680 | Invitrogen | #A27042 | WB (1:10,000), secondary |
| Antibody | Anti-rabbit biotinylated secondary antibody | Cell Signaling Technology | #14708 | Histology (1:500) |
| Chemical compound, drug | 5-hydroxytamoxifen | Sigma | T176 | Used in vitro |
| Commercial assay or kit | RNeasy kit | Qiagen Inc | #74104 | |
| Commercial assay or kit | EnzyChrom Triglyceride Assay Kit | BioAssay Systems | #ETGA-200 | |

*Continued on next page*

*Continued*

| Reagent type (species) or resource | Designation | Source or reference | Identifiers | Additional information |
|---|---|---|---|---|
| Commercial assay or kit | Mouse/rat fibroblast growth factor 21 ELISA | Biovendor | #RD291108200R | |
| Commercial assay or kit | Ultra-Sensitive Mouse Insulin ELISA Kit | Chrystal Chem | #90080 | |
| Commercial assay or kit | High-Capacity cDNA reverse Transcription Kit | Applied Biosystems | #4368814 | |
| Commercial assay or kit | Hematoxylin and Eosin Stain Kit | Vector Laboratories | #H3502 | |
| Software, algorithm | GraphPad Prism Software | GraphPad Software, La Jolla, CA, USA | Version 8.0.0 for Windows RRID:SCR_002798 | |
| Other | 2920X, standard chow | Harlan Teklad | 2920X | |
| Other | Chow, 60% HFD | Research Diets | D12492 | |
| Other | Chow, 10% Control | Research Diets | D12450J | |
| Sequence-based reagent | Fgf21_F | Integrated DNA Technologies, Inc | PCR primers | TGACGACCAAGACACTGAAGC |
| Sequence-based reagent | Fgf21_R | Integrated DNA Technologies, Inc | PCR primers | TTTGAGCTCCAGGAGACTTTCTG |
| Sequence-based reagent | Atf4_F | Integrated DNA Technologies, Inc | PCR primers | AGCAAAACAAGACAGCAGCC |
| Sequence-based reagent | Atf4_R | Integrated DNA Technologies, Inc | PCR primers | ACTCTCTTCTTCCCCCTTGC |
| Sequence-based reagent | Chop_F | Integrated DNA Technologies, Inc | PCR primers | GTCCCTAGCTTGGCTGACAGA |
| Sequence-based reagent | Chop _R | Integrated DNA Technologies, Inc | PCR primers | TGGAGAGCGAGGGCTTTG |
| Sequence-based reagent | Ern1_F | Integrated DNA Technologies, Inc | PCR primers | TGAAACACC CCTTCTTCTGG |
| Sequence-based reagent | Ern1_R | Integrated DNA Technologies, Inc | PCR primers | CCT CCT TTT CTA TTC GGT CAC TT |
| Sequence-based reagent | Opa1_F | Integrated DNA Technologies, Inc | PCR primers | ATACTGGGATCTGCTGTTGG |
| Sequence-based reagent | Opa1_R | Integrated DNA Technologies, Inc | PCR primers | AAGTCAGGCACAATCCACTT |
| Sequence-based reagent | Ucp1_F | Integrated DNA Technologies, Inc | PCR primers | GTGAAGGTCAGAATGCAAGC |
| Sequence-based reagent | Ucp1_R | Integrated DNA Technologies, Inc | PCR primers | AGGGCCCCCTTCATGAGGTC |
| Sequence-based reagent | Prdm16_F | Integrated DNA Technologies, Inc | PCR primers | CAGCACGGTGAAGCCATTC |
| Sequence-based reagent | Prdm16_R | Integrated DNA Technologies, Inc | PCR primers | GCGTGCATCCGCTTGTG |
| Sequence-based reagent | Gapdh_F | Integrated DNA Technologies, Inc | PCR primers | AACGACCCCTTCATTGAC |
| Sequence-based reagent | Gapdh_R | Integrated DNA Technologies, Inc | PCR primers | TCCACGACATACTCAGCAC |
| Sequence-based reagent | Ppargc1a_F | Integrated DNA Technologies, Inc | PCR primers | GTAAATCTGCGGGATGATGG |
| Sequence-based reagent | Ppargc1a_R | Integrated DNA Technologies, Inc | PCR primers | AGCAGGGTCAAAATCGTCTG |
| Sequence-based reagent | Dio2_F | Integrated DNA Technologies, Inc | PCR primers | AATTATGCCTCGGAGAAGACCG |

*Continued on next page*

*Continued*

| Reagent type (species) or resource | Designation | Source or reference | Identifiers | Additional information |
|---|---|---|---|---|
| Sequence-based reagent | *Dio2_R* | Integrated DNA Technologies, Inc | PCR primers | GGCAGTTGCCTAGTGAAAGGT |
| Sequence-based reagent | *Nrg4_F* | Integrated DNA Technologies, Inc | PCR primers | ACTCACTAAGCCAGAGTGAAGCAGG |
| Sequence-based reagent | *Nrg4_R* | Integrated DNA Technologies, Inc | PCR primers | CATGTCGTCTCTACAGGTGCTCTGC |
| Sequence-based reagent | *Cre_F* | Integrated DNA Technologies, Inc | PCR primers | AATGCTTCTGTCCGTTTGCC |
| Sequence-based reagent | *Cre_R* | Integrated DNA Technologies, Inc | PCR primers | ACATCTTCAGGTTCTGCGGG |
| Sequence-based reagent | *Cpt1b_F* | Integrated DNA Technologies, Inc | PCR primers | TGCCTTTACATCGTCTCCAA |
| Sequence-based reagent | *Cpt1b_R* | Integrated DNA Technologies, Inc | PCR primers | AGACCCCGTAGCCATCATC |
| Sequence-based reagent | *Ppara_F* | Integrated DNA Technologies, Inc | PCR primers | GAGAATCCACGAAGCCTACC |
| Sequence-based reagent | *Ppara_R* | Integrated DNA Technologies, Inc | PCR primers | ATTCGGACCTCTGCCTCTTT |
| Sequence-based reagent | *Acadm_F* | Integrated DNA Technologies, Inc | PCR primers | ACTGACGCCGTTCAGATTTT |
| Sequence-based reagent | *Acadm_R* | Integrated DNA Technologies, Inc | PCR primers | GCTTAGTTACACGAGGGTGATG |
| Sequence-based reagent | *Metrnl_F* | Integrated DNA Technologies, Inc | PCR primers | CTGGAGCAGGGAGGCTTATTT |
| Sequence-based reagent | *Metrnl_R* | Integrated DNA Technologies, Inc | PCR primers | GGACAACAAAGTCACTGGTACAG |
| Sequence-based reagent | *Bmp8b_F* | Integrated DNA Technologies, Inc | PCR primers | CAACCACGCCACTATGCA |
| Sequence-based reagent | *Bmp8b_R* | Integrated DNA Technologies, Inc | PCR primers | CACTCAGCTCAGTAGGCACA |
| Sequence-based reagent | *Slit2-c_F* | Integrated DNA Technologies, Inc | PCR primers | GCTGTGAACCATGCCACAAG |
| Sequence-based reagent | *Slilt2-c_R* | Integrated DNA Technologies, Inc | PCR primers | CACACATTTGTTTCCGAGGCA |
| Sequence-based reagent | *Evlov6_F* | Integrated DNA Technologies, Inc | PCR primers | TCAGCAAAGCACCCGAAC |
| Sequence-based reagent | *Evlov6_R* | Integrated DNA Technologies, Inc | PCR primers | AGCGACCATGTCTTTGTAGGAG |
| Sequence-based reagent | *Il6_F* | Integrated DNA Technologies, Inc | PCR primers | TGGGAAATCGTGGAAATGAG |
| Sequence-based reagent | *Il6_R* | Integrated DNA Technologies, Inc | PCR primers | GAAGGACTCTGGCTTTGTCTT |

## Mouse models

Experiments were performed in male and/or female mice on a C57Bl/6J background. *Opa1*fl/fl mice (*Zhang et al., 2011*), *Fgf21*fl/fl mice (*Potthoff et al., 2009*), and *Atf4*fl/fl mice were (*Ebert et al., 2012*) generated as previously described. Transgenic mice expressing cre recombinase under the control of the *Ucp1* promoter (Tg (*Ucp1*-cre)1Evdr) (*Kong et al., 2014*) and transgenic mice expressing a tamoxifen-inducible cre under the control of the *Adipoq* gene promoter (C57BL/6-Tg (*Adipoq*-cre/ERT2)1Soff/J) (*Sassmann et al., 2010*) were acquired from the Jackson Laboratories (#024670 and #025124, respectively). Mice were weaned at 3 weeks of age and were either kept on standard chow (2920X Harlan Teklad, Indianapolis, IN, USA) or were fed special diets. For DIO studies, 6-

week-old mice were divided into a control diet group (Cont; 10% kcal from fat—Research Diets, New Brunswick, NJ, USA, D12450J) or a HFD group (60% kcal from fat—Research Diets D12492) and were kept on these respective diets for 12 weeks. For the cold exposure experiments, mice were acclimated to 30°C (thermoneutral temperature for mice) for 7 days prior to being cold-exposed. Unless otherwise noted, animals were housed at 22°C with a 12 hr light, 12 hr dark cycle with free access to water and standard chow or special diets. All mouse experiments presented in this study were conducted in accordance with the animal research guidelines from NIH and were approved by the University of Iowa IACUC.

## Methods details

### Studies with mice reared at thermoneutrality

A subset of OPA1 BAT KO female mice and their WT littermate controls were transferred to a rodent environmental chamber (Power Scientific) set at 30°C following weaning (~4 weeks of age) and were housed for the subsequent 4 weeks (until ~8 weeks of age). Body composition and indirect calorimetry were measured at the end of the 4 weeks. Mice were transferred to an OxyMax Comprehensive Lab Animal Monitoring System (CLAMS, Columbus Instruments International), where oxygen consumption, food intake, and ambulatory activity were measured. Body composition was determined by NMR, and BAT, iWAT, and gWAT depots were weighed upon tissue harvest.

### Cold exposure experiments

Core body temperature telemeters (Respironics, G2 E-Mitter, Murrysville, PA, USA) were surgically implanted into the abdominal cavity of 8–10-week-old male mice, and mice were then allowed to recover for 6 days post-surgery, while individually housed in a rodent environmental chamber (Power Scientific) at 30°C. Mice were then transferred to an OxyMax Comprehensive Lab Animal Monitoring System (CLAMS, Columbus Instruments International) at 30°C for 4 days, followed by 4°C for 3 days, as previously described (*Fisher et al., 2012*). Core body temperature was recorded every 17 min throughout the experiment, along with $O_2$ and $CO_2$ levels, food intake, and ambulatory activity, as estimated by photoelectric beam breaks in the X + Y plane. For the acute cold exposure experiments, 8-week-old mice were initially individually housed in the rodent environmental chamber at 30°C for 7 days. The initial temperature (t0) was recorded using a rectal probe (Fisher Scientific, Lenexa, KS, USA) at 7 am on day 8, after which the temperature was switched to 4°C. Once the desired temperature was reached, we recorded rectal temperatures hourly for up to 4 hr of cold exposure.

### Sympathetic nerve recording

Each mouse was anesthetized with intraperitoneal administration of ketamine (91 mg/kg BW (body weight)) and xylazine (9.1 mg/kg BW (body weight)). With the mouse in the dorsal position, a tracheotomy was performed and a PE-50 tubing was inserted to provide an unimpeded airway for the mouse to spontaneously breathe $O_2$-enriched room air. Next, a tapered micro-renathane tubing (Braintree Scientific, MRE-40) was inserted into the right jugular vein for infusion of the sustaining anesthetic agent ($\alpha$-chloralose: initial dose of 12 mg/kg, then sustaining dose of 6 mg/kg/hr). A second tapered MRE-40 catheter was inserted into the left common carotid artery that was attached to a pressure transducer (iWorx Systems, Inc, BP-100) for continuous measurement of arterial pressure and heart rate. Core body temperature was monitored with a rectal probe and maintained throughout the experiment with a temperature controller (Physitemp, Model TCAT2) set at 37.5°C.

To gain access to the nerve fascicle that innervates the inguinal white adipose fat, a small dermal incision was performed between the lower abdominal area and the right hindlimb. A single multi-fiber inguinal nerve was isolated from the nearby white fat deposit and placed on a bipolar platinum-iridium electrode (A-M Systems, 36-gauge), secured with silicone gel (WPI, Kwik-Sil). The nerve was carefully sectioned distal to the site of the recording. The electrode was attached to a high-impedance probe (Grass Instruments, HIP-511), and the nerve signal was filtered at a 100- and 1000 Hz cutoff and amplified by $10^5$ times with a Grass P511 AC pre-amplifier. The nerve signal was routed to a speaker system and to an oscilloscope (Hewlett-Packard, model 54501A) to monitor the audio and visual quality of the nerve recording. The nerve signal was also directed to a resetting voltage integrator (University of Iowa Bioengineering, model B600c) to analyze the total activity

(integrated voltage) and finally to a MacLab analog-digital converter (ADInstruments, Castle Hill, New South Wales, Australia, Model 8S) containing the software (MacLab Chart Pro; version 7.0) that utilizes a cursor to count the number of spikes/second that exceed the background noise threshold. Under a stable plane of anesthesia and strict isothermal conditions (37.5°C), continuous recording of baseline efferent inguinal WAT SNA was measured over a 30 min period. At the conclusion, the afferent end of the inguinal nerve-electrode complex was cut and the residual background noise measured, which was subtracted from the measurements to determine the real efferent inguinal SNA.

## GTTs, ITTs, nuclear magnetic resonance, and serum analysis

GTTs were performed after a 6 hr fast, and mice were administered glucose intraperitoneally (2 g/kg body weight), as described (*Tabbi-Anneni et al., 2008*). ITTs were performed after a 2 hr fast by injecting insulin intraperitoneally (0.75 U/kg body weight; Humulin, Eli Lilly, Indianapolis, IN, USA). Blood glucose was determined using a glucometer at regular time intervals (Glucometer Elite; Bayer, Tarrytown, NY, USA). Insulin solution was prepared in sterile 0.9% saline and dosed based on body weight. Plasma insulin was measured after a 6 hr fast using a commercially available kit according to the manufacturer's directions (Ultra-Sensitive Mouse Insulin ELISA Kit, Chrystal Chem, Downers Grove, IL, USA). Serum FGF21 (BioVendor ELISA kit, Asheville, NC, USA) was measured using commercially available kits according to the manufacturer's directions. Whole-body composition was measured by nuclear magnetic resonance in the Bruker Minispec NF-50 instrument (Bruker, Billerica, MA, USA). NMR was performed at the University of Iowa Fraternal Order of Eagles Diabetes Research Center Metabolic Phenotyping Core.

## Analysis of triglyceride levels

Triglycerides levels were measured in liver and in serum collected after a 6 hr fast using the Enzy-Chrom Triglyceride Assay Kit (BioAssay Systems, Hayward, CA, USA). Liver triglycerides were extracted using a solution of isopropanol and Triton X-100, as recommended by the manufacturer (*Tam et al., 2010*).

## RNA extraction and quantitative RT–PCR

Total RNA was extracted from tissues with TRIzol reagent (Invitrogen) and purified with the RNeasy kit (Qiagen Inc, Germantown, MD, USA). RNA concentration was determined by measuring the absorbance at 260 and 280 nm using a spectrophotometer (NanoDrop 1000, NanoDrop products, Wilmington, DE, USA). Total RNA (1 μg) was reverse-transcribed using the High-Capacity cDNA Reverse Transcription Kit (Applied Biosystems, Waltham, MA, USA), followed by qPCR reactions using SYBR Green (Life Technologies, Carlsbad, CA, USA) (*Pereira et al., 2017*). Samples were loaded in a 384-well plate in triplicate, and real-time polymerase chain reaction was performed with an ABI Prism 7900HT instrument (Applied Biosystems). The following cycle profile was used: 1 cycle at 95°C for 10 min; 40 cycles of 95°C for 15 s; 59°C for 15 s, 72°C for 30 s, and 78°C for 10 s; 1 cycle of 95°C for 15 s; 1 cycle of 60°C for 15 s; and 1 cycle of 95°C for 15 s. Data were normalized to *Gapdh* expression, and results are shown as relative mRNA levels. qPCR primers were designed using Primer-Blast or previously published sequences (*Kim et al., 2013*).

## Western blot analysis

Immunoblotting analysis was performed as previously described (*Pereira et al., 2013*). Approximately 50 mg of frozen tissue was homogenized in 200 μl lysis buffer containing (in mmol/l) 50 HEPES, 150 NaCl, 10% glycerol, 1% Triton X-100, 1.5 $MgCl_2$, 1 EGTA, 10 sodium pyrophosphate, 100 sodium fluoride, and 100 μmol/l sodium vanadate. Right before use, HALT protease/phosphatase inhibitors (Thermo Fisher Scientific, Waltham, MA, USA) were added to the lysis buffer and samples were processed using the TissueLyser II (Qiagen Inc). Tissue lysates or freshly isolated mitochondria were resolved on SDS–PAGE and transferred to nitrocellulose membranes (Millipore Corp., Billerica, MA, USA). Membranes were incubated with primary antibodies overnight and with secondary antibodies for 1 hr at room temperature. Fluorescence was quantified using the LiCor Odyssey imager.

## Mitochondrial isolation

Mitochondrial fraction was isolated from iBAT or from iWAT, as previously described (*Garcia-Cazarin et al., 2011*). Briefly, tissue was excised, rinsed in ice-cold PBS, and maintained in ice-cold isolation buffer (500 mM EDTA, 215 mM D-mannitol, 75 mM sucrose, 0.1% free fatty acid bovine serum albumin, 20 mM HEPES, pH 7.4 with KOH) until ready for homogenization. Bradford assay was performed to determine the protein concentration.

## Oxygen consumption

Mitochondrial function was assessed using Seahorse XF96 analyzer. Briefly, 2.5 µg of BAT or iWAT mitochondria were seeded on a polyethylene terephthalate (PET) plate and centrifuged for 20 min at $2000 \times g$ at 4°C. Substrates were added to the assay buffer at the following final concentrations: pyruvate at 2 mM, malate at 0.8 mM, and palmitoyl carnitine at 0.02 mM. Measurements were performed at baseline conditions (state 2) or after ADP (2 mM) injection (state 3). Substrates were freshly prepared, and reagents were purchased from Sigma (St. Louis, MO, USA).

## ATP synthesis rates

ATP synthesis rates were assessed in 20 µg of mitochondria isolated from iBAT using a fluorometer (Horiba Systems, Irvine, CA, USA). Briefly, buffer Z lite (105 mM KMES, 30 mM KCl, 10 mM $KH_2PO_4$, 5 mM $MgCl_2.6H_2O$, 0.5 mg/ml BSA, pH 7.4 with KOH) supplemented with glucose, hexokinase/G-6-PDH (Sigma), Ap5a (an inhibitor of adenylate kinase, Sigma), and NADP (Sigma) was added with 20 µg of mitochondria. ADP (75 µM)-stimulated ATP synthesis rates were measured kinetically in the presence of palmitoyl carnitine (5 µM) and malate (1.6 mM) from the formation of NADPH in a coupled reaction that uses ATP for conversion of glucose to glucose-6-phosphate (G6P) by hexokinase and subsequently to 6-phosphogluconolactone by G6P dehydrogenase coupled with reduction of NADP to NADPH. NADPH accumulation is measured by fluorometry using excitation/emission wavelengths of 345 nm/460 nm, respectively (*Lark et al., 2016*).

## Transmission electron microscopy

Electron micrographs of iBAT and iWAT were prepared as previously described (*Pereira et al., 2017*). Briefly, iBAT and iWAT were trimmed into tiny pieces using a new blade to minimize mechanical trauma to the tissue. Tissues were fixed overnight (in 2% formaldehyde and 2.5% glutaraldehyde), rinsed (0.1% cacodylate pH 7.2) and stained with increasing concentrations of osmium (1.5%, 4%, 6%), and dehydrated with increasing concentrations of acetone (50%, 75%, 95%, 100%). Samples were then embedded, cured, sectioned, and poststained with uranyl and lead. Sections were then imaged on a Jeol 1230 Transmission electron microscope.

## Histology and immunohistochemistry

Fragments of BAT and iWAT were embedded in paraffin, portioned into 5-µm-thick sections, and stained with hematoxylin-eosin (Fisher, Pittsburgh, PA, USA). For immunohistochemistry, iWAT sections were deparaffinized, re-hydrated, blocked with 10% goat serum, and incubated overnight with a rabbit primary antibody against UCP1 (Abcam, Ab1098, 1:250). Sections were, then, incubated with an anti-rabbit biotinylated secondary antibody (1:500) for 1 hr, incubated with peroxidase streptavidin solution (1:500) for 30 min, and revealed with DAB chromogen solution for 10 s (Vector Laboratories, Burlingame, CA, USA). Light microscopy was performed using a Nikon Eclipse Ti-S microscope (Nikon, Melville, NY, USA).

## Cell culture and treatments

BAT stromal vasculature fraction was isolated from 6-day-old *Opa1* floxed mice harboring the tamoxifen-inducible cre recombinase, Cre-ERT2, under the control of the *Adipoq* gene promoter. Cells were grown and differentiated as previously described (*BonDurant et al., 2017*). After cells were fully differentiated, brown adipocytes were treated with 500 nM 4-hydroxytamoxifen (Sigma) for 72 hr to induce *Opa1* deletion (KO cells) or with vehicle solution (WT cells). Cells were then switched to serum-free and phenol-free DMEM/F12 for 6 hr, after which the media was collected for FGF21 measurements. Cells were washed with ice-cold PBS and harvested for subsequent analysis.

## Data analysis

Unless otherwise noted, all data are reported as mean ± SEM. To determine statistical differences, Student's *t*-test was performed for comparison of two groups, and two-way ANOVA followed by Tukey multiple comparison test was utilized when more than three groups were compared. A probability value of $p \leq 0.05$ was considered significantly different. Statistical calculations were performed using the GraphPad Prism software (La Jolla, CA, USA). The association between oxygen consumption and body mass was calculated by ANCOVA, using the CalR software (*Mina et al., 2018*). The significance test for the 'group effect' determined whether the two groups of interest were significantly different for the metabolic variable selected. The significance test for 'mass effect' informs if there is an association between the mass variable and oxygen consumption among all animals in the study.

## Acknowledgements

This work was supported by grants HL127764 and HL112413 from the NIH, 20SFRN35120123 from the American Heart Association (AHA), and the Teresa Benoit Diabetes research fund to EDA, who is an established investigator of the AHA; by AHA Scientist Development Grant 15SDG25710438 and NIH DK125405 to ROP; by the Diabetes Research Training Program funded by the NIH (T32DK112751-01) to SHB; and by the NIH 1R25GM116686 to LMGP. Metabolic phenotyping was performed at the Metabolic Phenotyping Core at the Fraternal Order of Eagles Diabetes Research Center. Analysis of mRNA expression was performed by qPCR at the Genomics Division of The Iowa Institute of Human Genetics. Electron Microscopy was performed at the Central Microscopy Research Facility at the University of Iowa. We would like to thank Dr. Hiromi Sesaki, at John Hopkins, for providing us with the OPA1 floxed mice.

## Additional information

### Funding

| Funder | Grant reference number | Author |
|---|---|---|
| National Institutes of Health | HL127764 | E Dale Abel |
| National Institutes of Health | HL112413 | E Dale Abel |
| American Heart Association | 20SFRN35120123 | E Dale Abel |
| American Heart Association | 15SDG25710438 | Renata O Pereira |
| National Institutes of Health | DK125405 | Renata O Pereira |
| National Institutes of Health | T32DK112751 | Sarah Hartwick Bjorkman |
| National Institutes of Health | 1R25GM116686 | Luis Miguel García-Peña |

The funders had no role in study design, data collection and interpretation, or the decision to submit the work for publication.

### Author contributions

Renata O Pereira, Conceptualization, Data curation, Formal analysis, Supervision, Funding acquisition, Investigation, Methodology, Writing - original draft, Project administration, Writing - review and editing; Alex Marti, Angela Crystal Olvera, Satya Murthy Tadinada, Sarah Hartwick Bjorkman, Eric Thomas Weatherford, Donald A Morgan, Rhonda A Souvenir, Data curation, Formal analysis, Investigation, Methodology; Michael Westphal, Luis Miguel García-Peña, Data curation, Formal analysis, Investigation; Pooja H Patel, Ana Karina Kirby, Rana Hewezi, William Bùi Trân, Data curation, Investigation; Monika Mittal, Data curation, Formal analysis; Christopher M Adams, Matthew J Potthoff, Resources, Writing - review and editing; Kamal Rahmouni, Resources, Formal analysis, Investigation; E Dale Abel, Conceptualization, Resources, Supervision, Funding acquisition, Project administration, Writing - review and editing

## Author ORCIDs

Renata O Pereira (iD) https://orcid.org/0000-0001-5809-4669
Pooja H Patel (iD) https://orcid.org/0000-0002-5345-0158
Luis Miguel García-Peña (iD) http://orcid.org/0000-0001-8718-6490
Rhonda A Souvenir (iD) http://orcid.org/0000-0002-8880-2483
E Dale Abel (iD) https://orcid.org/0000-0001-5290-0738

## Ethics

Animal experimentation: This study was performed in strict accordance with the recommendations in the Guide for the Care and Use of Laboratory Animals of the National Institutes of Health. All of the animals were handled according to approved institutional animal care and use committee (IACUC) protocols (#8051408 and 0032294) of the University of Iowa.

## Decision letter and Author response

Decision letter https://doi.org/10.7554/eLife.66519.sa1
Author response https://doi.org/10.7554/eLife.66519.sa2

# Additional files

## Supplementary files

• Transparent reporting form

## Data availability

All data generated or analyzed during this study are included in the manuscript and supporting files. Source data files have been provided for all figures.

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
