## [Decision Letter]

**Acceptance summary:**

This manuscript represents an exciting advance because an animal model was adopted that displayed defective brown fat responses to cold exposure but were protected from diet-induced obesity. The authors show that this may be due to release of FGF21 from the brown adipocytes. However, brown adipocyte-specific deletion of both OPA1 and FGF21 did not support this hypothesis. The authors propose that deletion of OPA1 leads to endoplasmic reticulum stress, which activates the transcriptional mediator ATF4. Deleting OPA1 and ATF4 in brown adipocytes largely phenocopies the phenotype seen with deletion of OPA1 and FGF21. The authors went on to perform sympathetic nerve recordings which support these observations. These studies provide novel insights into inter-tissue cross talk and metabolic homeostasis.

**Decision letter after peer review:**

Thank you for submitting your article "OPA1 Deletion in Brown Adipose Tissue Improves Thermoregulation and Systemic Metabolism via FGF21" for consideration by *eLife*. Your article has been reviewed by 3 peer reviewers, one of whom is a member of our Board of Reviewing Editors, and the evaluation has been overseen by David James as the Senior Editor. The reviewers have opted to remain anonymous.

Essential revisions:

1. In Figure 3I it is demonstrated that serum levels of FGF21 are induced in the BAT-OPA1 knockout animals. Close inspection of the data reveals some variability in the serum FGF21 levels within the KO animals. 4 animals have induced levels of serum FGF21 whereas ~12 animals have levels within the range found in control mice. The authors should discuss this. Does the level of compensatory FGF21 induction correlate with other parameters in individual mice?

2. The authors show whole-body energy expenditure data in several figures (Figure 1, 2, 4, 5, 6, and 7). However, the VO2 data are distilled into bar graphs normalized to body mass. This means of displaying the data is not the standard in the field and has the potential to lead to erroneous interpretations. This is particularly the case when comparing animals of different body weight and body composition. The authors should refer to Tschop et al. (Nat Methods, 2012, 9: 57-63) and Mina et al. (Cell Metab, 2018, 28: 656-66) to ensure their data is analyzed and displayed in a standard fashion.

3. It would be appropriate to include levels of UCP1 as part of the analysis in Figure 1.

4. Western blots should include some molecular markers.

5. When does the phenotype emerge? The earliest time point examined appears to be 8 weeks of age.

6. Despite providing a wealth of new models and an abundance of data, this manuscript is lacking in mechanistic underpinnings. Moreover, the authors make claims regarding possible mechanisms that are not well supported by the data. Specifically, in explaining the browning of white fat seen in their models, the authors rely on Western blots for TH in whole adipose tissue to argue for increased sympathetic innervation of iWAT. In addition to being generally unreliable (TH is also expressed in some sensory neurons), to justify this claim additional experiments would be needed, such as (a) morphology: immunohistochemistry and/or whole mount imaging, (b) function: sympathetic nerve recordings, mass spec or HPLC measurement of tissue catecholamines, and (c) downstream effects: assays of lipolysis and/or signaling mediators downstream of sympathetic activation, such as pHSL vs. total HSL. These issues and limitations should be discussed in the text.

---

## [Author Response]

Essential revisions:1. In Figure 3I it is demonstrated that serum levels of FGF21 are induced in the BAT-OPA1 knockout animals. Close inspection of the data reveals some variability in the serum FGF21 levels within the KO animals. 4 animals have induced levels of serum FGF21 whereas ~12 animals have levels within the range found in control mice. The authors should discuss this. Does the level of compensatory FGF21 induction correlate with other parameters in individual mice?

We appreciate your comment on this matter. Unfortunately, because the serum was collected under ad libitum feeding conditions (between 8 am-12 pm), variability in the data depicted in Figure 3I (now Figure 3J of the revised manuscript) could reflect differences in feeding status in random-fed animals. To control for this possibility, we measured FGF21 levels in a separate cohort of animals after a 6-hour fast (Figure S3B). Although some variability remains, values for FGF21 in KO animals do not overlap with any values in control mice. These data confirm an almost 2-fold increase in FGF21 circulating levels in KO mice. Generally speaking, comparison of the data in Figure 3J and S3B indicate that FGF21 levels are lower in WT and KO mice in the random-fed state. Interestingly, the FGF21 levels in the 4 KO animals highlighted by the reviewers, are close to those of WT mice that were fasted for 6 hours, which supports the impact of the feeding state of animals to confound differences between genotypes. Importantly, even if the 4 animals with the greatest degree of FGF21 induction are removed from the analysis in Figure 3l, the increase in circulating FGF21 in the KO group, although attenuated remains statistically significantly increased. Therefore, we conclude that the variability in FGF21 circulating levels in random-fed animals is most likely related to differences in feeding status at the time of serum collection.

2. The authors show whole-body energy expenditure data in several figures (Figure 1, 2, 4, 5, 6, and 7). However, the VO2 data are distilled into bar graphs normalized to body mass. This means of displaying the data is not the standard in the field and has the potential to lead to erroneous interpretations. This is particularly the case when comparing animals of different body weight and body composition. The authors should refer to Tschop et al. (Nat Methods, 2012, 9: 57-63) and Mina et al. (Cell Metab, 2018, 28: 656-66) to ensure their data is analyzed and displayed in a standard fashion.

Thank you for highlighting this. We have now analyzed our data using CalR, the web-based tool for analysis of experiments using indirect calorimetry to measure physiological energy balance. ANCOVA is now used to compare differences in oxygen consumption, with genotype as the categorical independent variable and body mass as the dependent variable.

3. It would be appropriate to include levels of UCP1 as part of the analysis in Figure 1.

We have added UCP1 protein levels to figure 1F, alongside OPA1 levels in BAT.

4. Western blots should include some molecular markers.

We have included molecular weight markers for the specific proteins measured in the manuscript.

5. When does the phenotype emerge? The earliest time point examined appears to be 8 weeks of age.

We have data in 5-week-old females that are now included in Figure S1. At this time point, we did not detect any changes in body mass or body composition. By 7 weeks, females have increased oxygen consumption, in the absence of significant differences in body mass. These additional data suggest that the earliest time-point reported in the main manuscript (8 weeks) likely represent a time point when most phenotypical changes emerge between genotypes. It also strengthens our conclusion that an increase in basal metabolic rates likely contributes to the lean phenotype in OPA1 BAT KO mice.

6. Despite providing a wealth of new models and an abundance of data, this manuscript is lacking in mechanistic underpinnings. Moreover, the authors make claims regarding possible mechanisms that are not well supported by the data. Specifically, in explaining the browning of white fat seen in their models, the authors rely on Western blots for TH in whole adipose tissue to argue for increased sympathetic innervation of iWAT. In addition to being generally unreliable (TH is also expressed in some sensory neurons), to justify this claim additional experiments would be needed, such as (a) morphology: immunohistochemistry and/or whole mount imaging, (b) function: sympathetic nerve recordings, mass spec or HPLC measurement of tissue catecholamines, and (c) downstream effects: assays of lipolysis and/or signaling mediators downstream of sympathetic activation, such as pHSL vs. total HSL. These issues and limitations should be discussed in the text.

Thank you for these comments. To provide additional mechanism to support increased sympathetic innervation of iWAT we performed sympathetic nerve (SNA) recordings for nerves that innervate iWAT of WT and KO mice (Figure 3H). These SNA recordings confirm increased sympathetic activity in iWAT of KO mice. We also discuss potential limitations of these data and additional confirmatory assays in our Discussion section, as we do not have nerve recordings for all the different mouse models included in the manuscript.